# Confined migration promotes cancer metastasis through resistance to anoikis and increased invasiveness

Deborah Fanfone[1,2], Zhichong Wu[1,3,4,5], Jade Mammi[1,2], Kevin Berthenet[1,2,4], David Neves[6], Kathrin Weber[1,2], Andrea Halaburkova[1,2], François Virard[1,7], Félix Bunel[8], Catherine Jamard[1,2], Hector Hernandez-Vargas[1,4,9], Stephen WG Tait[10,11], Ana Hennino[1,3,4], Gabriel Ichim[1,2]*

[1]Cancer Research Center of Lyon (CRCL), INSERM 1052, CNRS, Lyon, France; [2]Cancer Cell Death Laboratory, part of LabEx DEVweCAN, Université de Lyon, Lyon, France; [3]Université Lyon 1, Villeurbanne, Villeurbanne, France; [4]Centre Léon Bérard, Lyon, France; [5]Department of General Surgery, Pancreatic Disease Center, Ruijin Hospital, Shanghai Jiao Tong University School of Medicine, Shanghai, China; [6]Netris Pharma, Lyon, France; [7]Université Claude Bernard Lyon 1, Faculté d'Odontologie, Hospices Civils de Lyon, Lyon, France; [8]ENS de Lyon, Université Claude Bernard Lyon 1, CNRS, Laboratoire de Physique, Lyon, France; [9]Université Claude Bernard Lyon 1, Lyon, France; [10]Cancer Research UK Beatson Institute, Glasgow, United Kingdom; [11]Institute of Cancer Sciences, University of Glasgow, Glasgow, United Kingdom

*For correspondence:
gabriel.ichim@lyon.unicancer.fr

**Competing interest:** The authors declare that no competing interests exist.

**Abstract** Mechanical stress is known to fuel several hallmarks of cancer, ranging from genome instability to uncontrolled proliferation or invasion. Cancer cells are constantly challenged by mechanical stresses not only in the primary tumour but also during metastasis. However, this latter has seldom been studied with regards to mechanobiology, in particular resistance to anoikis, a cell death programme triggered by loss of cell adhesion. Here, we show in vitro that migrating breast cancer cells develop resistance to anoikis following their passage through microporous membranes mimicking confined migration (CM), a mechanical constriction that cancer cells encounter during metastasis. This CM-induced resistance was mediated by Inhibitory of Apoptosis Proteins, and sensitivity to anoikis could be restored after their inhibition using second mitochondria-derived activator of caspase (SMAC) mimetics. Anoikis-resistant mechanically stressed cancer cells displayed enhanced cell motility and evasion from natural killer cell-mediated immune surveillance, as well as a marked advantage to form lung metastatic lesions in mice. Our findings reveal that CM increases the metastatic potential of breast cancer cells.

## Editor's evaluation

The authors provide data supporting the notion that while cells expressing higher levels of cIAP1 or *XIAP* have no confined migratory advantage, these apoptosis inhibitors are upregulated in response to confined migration, which provides cells with a migratory and survival (e.g., anoikis resistance) advantage as well as means to evade NK cells, resulting in increased metastasis.

## Introduction

The majority of cancer-related deaths arise following metastasis (*Dillekås et al., 2019*; *Chaffer and Weinberg, 2011*). Metastatic cancer cells acquire de novo phenotypic traits allowing them to efficiently leave the primary tumour, enter blood circulation, and survive harsh conditions, then exit the bloodstream and establish metastasis at a distant site. Though the mechanisms driving cancer cell migration and invasion are well documented, with a clear understanding of epithelial–mesenchymal transition and the metastatic niche, no efficient therapeutic strategies currently prevent metastasis formation (*Lambert et al., 2017*). Metastasis is thus largely incurable, yet this lengthy process can take years and the therapeutic window is therefore large enough to envisage its targeting (*Mehlen and Puisieux, 2006*).

Multiple mechanical forces occur at each step of cancer development, from the primary tumour to metastasis. During early tumour development, excessive cell proliferation, massive extracellular matrix deposition, or cancer-associated fibroblasts exert compressive forces on cancer cells that can reach up to 10kPa in pancreatic ductal adenocarcinoma (*Nia et al., 2016*). As they engage in their metastatic journey, cancer cells migrate through barriers including desmoplastic tumour stroma, basal membranes, endothelial layers, and when entering low-diameter capillaries. In healthy tissues or tumours, the extracellular matrix creates pores or tunnel-shaped tracks that are often smaller than the diameter of a cell and migrating cells adjust their shape and size according to these constrictions (*Butcher et al., 2009*). These events, known as confined migration (CM), have dramatic consequences on cancer cells and might even break the nuclear envelope and trigger DNA damage and mutagenesis if the breaks are not efficiently repaired (*Raab et al., 2016*; *Denais et al., 2016*).

CM can alter the phenotypic traits of cancer cells rendering them even more aggressive. More precisely, repeated nuclear deformations and loss of nuclear envelope integrity can activate an invasive programme, and engage the pro-oncogenic Ras/MAPK signalling pathway (*Nader et al., 2021*; *Rudzka et al., 2021*; *Rudzka et al., 2019*). Once inside the blood or the lymphatic vessels, circulating tumour cells (CTCs) are confronted with deadly fluid shear stress until they extravasate (*Fan et al., 2016*). Moreover, most CTCs are eliminated by apoptosis in a process called anoikis, occurring when cells detach from the extracellular matrix (*Shen and Kang, 2020*; *Paoli et al., 2013*). By rapidly engaging either the death receptors or the mitochondrial pathway of apoptosis, anoikis has evolved as an efficient physiological barrier for preventing the formation of metastatic colonies by CTCs reaching target organs (*Shen and Kang, 2020*). Nonetheless, cancer cells developed strategies to evade anoikis such as overly activated Ras/ERK and PI3K/Akt pathways, engaging the tyrosine kinase receptor TrkB, inactivation of E-cadherin and p53 or enhanced autophagy (*de Sousa Mesquita et al., 2017*; *Douma et al., 2004*; *Derksen et al., 2006*; *Fung et al., 2008*; *Chavez-Dominguez et al., 2020*).

Mechanical stress has emerged as a key factor in shaping the pro-metastatic features of cancer cells. We thus hypothesized that CM may also contribute to the metastatic potential by impacting anoikis and cancer invasiveness. We show here that breast cancer cells having undergone CM, but not compression, become resistant to anoikis, through a mechanism involving lowering apoptotic caspase activation through an upregulation of Inhibitory of Apoptosis Proteins (IAPs). We also report that treatment with SMAC mimetics to lower IAPs expression restores the sensitivity to anoikis. Ultimately, a single round of CM is sufficient to enhance emerging aggressiveness, the most obvious effects observed being for single-cell migration and escape from natural killer (NK) cell-mediated immune surveillance. In addition, these observations are endorsed in vivo by higher lung metastatic burden when mice are engrafted with breast cancer cells challenged by CM. Taken together, our results support that CM triggers a particular signalling signature that might favour certain metastatic hallmarks such as resistance to anoikis and increased invasiveness.

## Results

### CM confers breast cancer cells with resistance to anoikis

The human breast cancer cells MDA-MB-231 are highly invasive and aggressive in vitro and in vivo, and represent an ideal cellular model to study metastasis (*Cailleau et al., 1974*). To investigate the effects of constriction on these cells, we subjected them to a forced passage through a membrane with 3 µm in diameter pores, via a serum gradient, mimicking the CM encountered during cancer

progression (*Rudzka et al., 2021*; *Porporato et al., 2014*; *Xia et al., 2019*). MDA-MB-231 cells were seeded onto a matrigel-coated tissue culture insert, prior to applying the serum gradient, and thus initially invaded the matrigel plug before following the serum through the microporous membrane (*Figure 1A*). We ascertained that CM did not affect cell viability, by verifying the incorporation of Calcein AM (viability dye), and by showing that apoptosis-triggering cytochrome *c* was not released by the mitochondria of CM cells, as it co-localized with COX IV (mitochondrial marker) (*Figure 1B, C*). Since caspases are the main apoptotic executioners, we next tested if they were activated in CM-challenged cancer cells. For this experiment, MDA-MB-231 cells expressing a bimolecular fluorescence complementation-based caspase-3 reporter (*Zhang et al., 2013*), which is functional in actinomycin D-treated and not in CRISPR[BAX/BAK] cells, were subjected to CM (*Figure 1—figure supplement 1A, B*). Recovered cells were viable and had not activated apoptotic effector caspases (*Figure 1D*), substantiating our previous result. In a complementary approach, we determined cell cycle distribution and found that CM cells have a negligible proportion of cells in subG1 (apoptotic cells) right after constriction (*Figure 1—figure supplement 1C, D*) and while they have a slight arrest in G1 phase (*Figure 1—figure supplement 1E*), this does not affect the overall cellular proliferation (*Figure 1E*). In line with this, CM cells displayed an unaltered mitochondrial membrane potential, ATP production and did not generate excessive reactive oxygen species (ROS) (*Figure 1—figure supplement 1F–I*). Hence, the CM model used here did not alter cell viability and was deemed suitable for studying phenotypic changes occurring in these mechanically challenged cancer cells.

We then focused on their response to anoikis, as we hypothesized that circulating tumour cells (CTCs) capable of withstanding such a physiological barrier and forming metastases may have acquired tumourigenic properties through the unique mechanical constriction imposed by CM (*Shen and Kang, 2020*). Importantly, CM-challenged MDA-MB-231 cells survived, grew, and formed clonogenic structures in low attachment conditions, as evidenced by the spheres generated, more efficiently than control cells (*Figure 1F*). The effect was not restricted to MDA-MB-231 cells since CM-challenged Hs578T breast cancer cells also developed more colonies than control counterparts, indicating that CM-challenged cells had overcome anoikis (*Figure 1—figure supplement 1J*). This is also the case when cells are grown in soft agar, in an anchorage-independent manner (*Figure 1G*). In addition, resistance to anoikis was assessed by IncuCyte Imager-based real-time imaging, using SYTOX Green dye exclusion, and again breast cancer cells undergoing CM had a survival advantage when grown in low attachment conditions (*Figure 1H, I*). Next, we investigated whether the survival advantage acquired following a single round of CM was transient. MDA-MB-231 cells were challenged by CM once and anoikis resistance was quantified at 3, 5, and 7 days post-CM, revealing that resistance to anoikis was transient (*Figure 1J*). Since anoikis is a variant of apoptosis, we wondered whether the activation of pro-apoptotic effector caspases was affected by CM. This was assessed by immunoblotting for cleaved caspase-3 and PARP-1, a proxy for efficient caspase activation. Strikingly, CM-challenged cancer cells had lower caspase-3 processing into the active p17 and p19 fragments, whereas PARP-1 cleavage followed the same pattern (*Figure 1K, L*). Lower effector caspase activation was also confirmed using a fluorometric caspase-3/7 assay, demonstrating that inhibition was particularly important for cells grown in ultra-low attachment and soft agar conditions (further designated as anoikis-favouring conditions) (*Figure 1M*).

To verify whether this resistance to anoikis was specific to cells having undergone CM and could not arise following the compressive stress experienced within primary tumours upon uncontrolled proliferation or increased extracellular matrix deposition, we tested the effects of compression on resistance to anoikis. We exposed MDA-MB-231 cells in vitro to a defined compression by pressing them against a permeable membrane with a weighted piston. The different weights translated into different pressures (200, 400, or 600 Pa) (*Figure 1—figure supplement 1K*). As shown by the increased nuclear size in compressed cells, this device was suitable to evaluate the effects of compression (*Figure 1—figure supplement 1L*). Resistance to anoikis, as assessed by growing these cells in anoikis-favouring conditions, was not modified under compression (*Figure 1N, O*), suggesting that acquisition of resistance to anoikis may be specific to CM. In addition, cancer cell migration through 8 μm in diameter microporous transwells, which does not impose cellular constriction, did not confer cancer cells with resistance to anoikis (*Figure 1—figure supplement 1M, N*) and had no impact on caspase activation (*Figure 1—figure supplement 1O*). Of note, the expression of several anti-apoptotic proteins such as BCL-xL, BCL2, and MCL1 was unchanged in CM cells, suggesting another resistance mechanism to

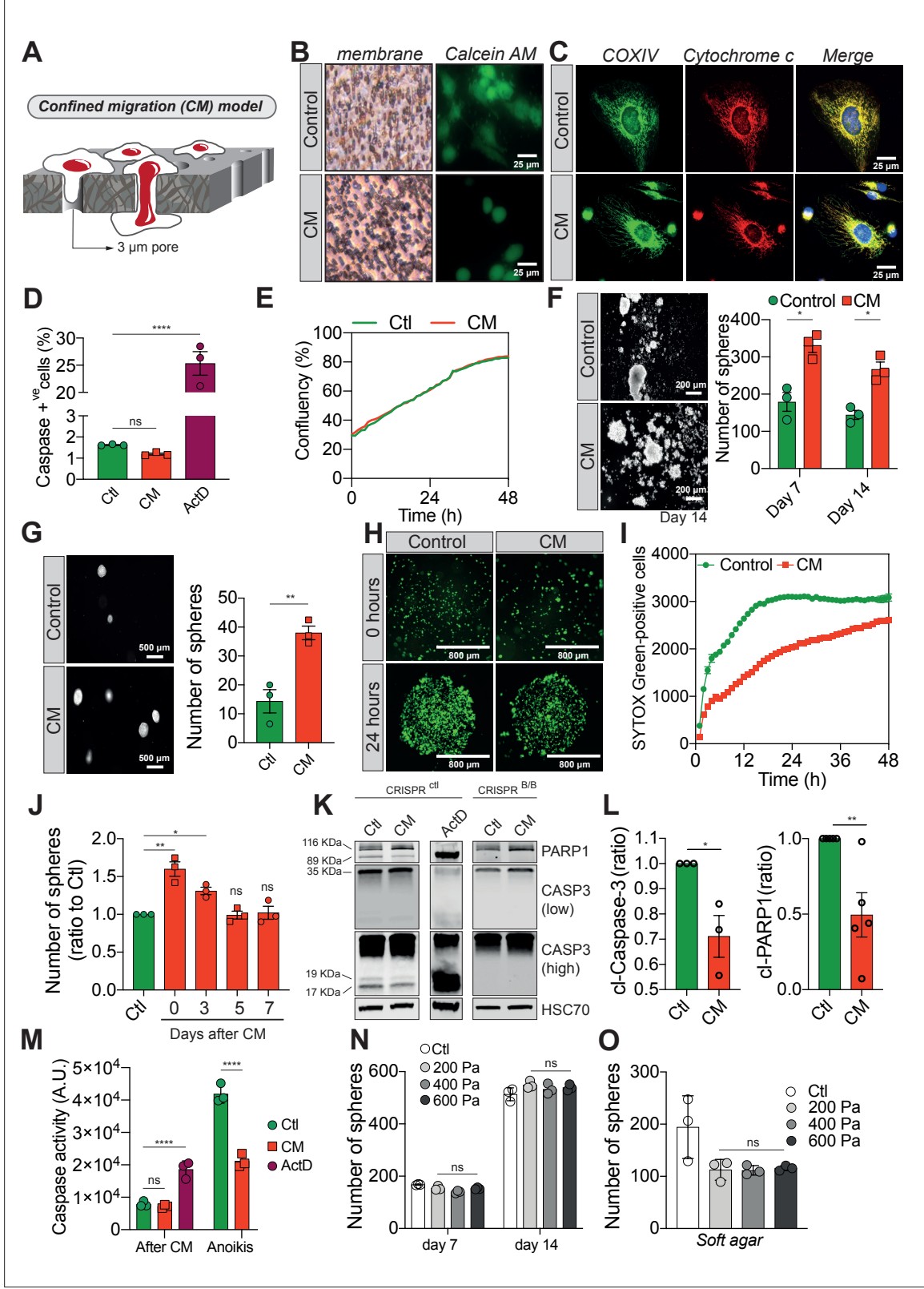

**Figure 1.** Confined migration (CM) confers breast cancer cells with resistance to anoikis. (**A**) Schematic illustration of the 3-μm transwell-based CM model. (**B**) MDA-MB-231 control cells or cells recovered from CM were stained with the Calcein AM viability dye and imaged by epifluorescence microscopy. (**C**) Representative immunofluorescence images of control and CM-challenged MDA-MB-231 cells stained for COX IV and cytochrome *c*. (**D**) Flow cytometry-based quantitative analysis of cells activating the VC3AI caspase reporter (*n* = 3, one-way analysis of variance [ANOVA] statistical

*Figure 1 continued on next page*

*Figure 1 continued*

test). (**E**) IncuCyte ZOOM live-cell imaging-based analysis of cell proliferation (*n* = 3, a representative experiment is shown). (**F**) Representative images of clonogenic structures from control and confined MDA-MB-231 cells grown in anoikis-promoting, ultra-low attachment conditions (left panel). Corresponding quantification of resistance to anoikis after 7 and 14 days of culture (right panel, *n* = 3, two-way ANOVA statistical test). (**G**) Control and CM-challenged MDA-MB-231 cells were grown in soft agar to test their anchorage-independent growth. Left panel depicts representative images, while the right panel is the quantification of clonogenic structures after 1 month (*n* = 3, *t*-test). (**H**) Control and CM-challenged MDA-MB-231 cells were imaged for 48 hr in an IncuCyte ZOOM imager in anoikis-promoting, ultra-low attachment conditions, and stained with SYTOX Green (*n* = 3, a representative experiment is shown). (**I**) IncuCyte-based SYTOX Green staining quantification of cell survival in control and CM MDA-MB-231 cells in anoikis-promoting conditions (*n* = 3, a representative experiment is shown). (**J**) Quantitative analysis of clonogenic structures illustrating the duration of resistance to anoikis between control and CM MDA-MB-231 cells up to 7 days post-CM (*n* = 3, one-way ANOVA statistical test). (**K**) Western blot analysis of PARP-1 cleavage and caspase-3 processing following CM and anoikis growth in control and in CRISPR/Cas9-mediated BAX/BAK DKO MDA-MB-231 cells. Actinomycin D treatment (1 μM for 12 hr) is used as a positive control for induction of apoptosis. (**L**) Densitometry analysis of PARP-1 and caspase-3 cleavage (ratio of CM cells to control) in anoikis conditions in MDA-MB-231 cells (*n* = 3–4, *t*-test). (**M**) The effect of CM on effector caspase activation was assessed using a fluorometric assay. Caspase activation was tested either immediately after CM, or in cells that were subsequently grown 24 hr in anoikis conditions (*n* = 3, two-way ANOVA statistical test). (**N**) Quantification of clonogenic structures formed by control or compressed MDA-MB-231 cells (subjected to 200, 400, or 600 Pa of compression for 16 hr) after 7 and 14 days of culture in anoikis conditions (*n* = 3, two-way ANOVA statistical test). (**O**) Quantification of clonogenic structures formed in soft agar by compressed MDA-MB-231 cells (*n* = 3, one-way ANOVA statistical test). (Statistical significance: ns - P > 0.05; * - P ≤ 0.05; ** - P ≤ 0.01; *** - P ≤ 0.001; **** - P ≤ 0.0001).

The online version of this article includes the following figure supplement(s) for figure 1:

**Figure supplement 1.** Confined migration (CM) confers breast cancer cells with resistance to anoikis.

anoikis (*Figure 1—figure supplement 1P*). In conclusion, these results show that CM has a profound impact on cancer cells resistance to cell death through inhibition of pro-apoptotic caspases.

## CM-driven resistance to anoikis relies on the anti-apoptotic IAP proteins

Previous studies reported that IAPs such as cIAP1, cIAP2, and XIAP promote resistance to anoikis in several cancers, through caspase inhibition (*Toruner et al., 2006*; *Liu et al., 2006*; *Berezovskaya et al., 2005*). We therefore hypothesized that under CM challenge, IAPs expression may underlie resistance to anoikis. This was tested by immunoblotting for cIAP1, cIAP2, and XIAP protein expression in CM MDA-MB-231 cells grown in anoikis-favouring conditions, which revealed an upregulation of all three IAPs (*Figure 2A*, left panel for densitometry analysis). Conversely, MDA-MB-231 cells subjected to compression or migration through 8 μm in diameter microporous transwells had unaltered levels of IAPs (*Figure 2—figure supplement 1A, B*). To further investigate this correlation, we transiently overexpressed all three IAPs in MDA-MB-231 cells (*Figure 2B*). Enforced expression of cIAP1 and XIAP significantly enhanced resistance to anoikis in cells grown under anoikis-favouring conditions, while cIAP2 overexpression was dispensable (*Figure 2C–E*). In a complementary approach, we deleted all three IAPs in MDA-MB-231 cells through CRISPR/Cas9-mediated gene editing (*Figure 2F*). Following CM and growth in anoikis-favouring conditions, control cells (EV) displayed the expected resistance to anoikis, whereas IAP-depleted cells lost their survival advantage (*Figure 2G* and *Figure 2—figure supplement 1C*). In addition, the use of a SMAC mimetic, one of several developed to specifically induce IAP degradation, namely BV6, successfully depleted both cIAP1 and XIAP (*Figure 2H*; *Varfolomeev et al., 2007*). Interestingly, it also abrogated the resistance to anoikis observed in CM-challenged breast cancer cells (*Figure 2I, J*).

Next, we sought to understand the mechanisms responsible for IAPs upregulation in constricted cells. We initially performed qRT-PCR analysis to assess IAPs mRNA expression in CM cells and uncovered that their transcript expression was not increased compared to control cells, with a significant inhibition observed for cIAP2 and XIAP (*Figure 2—figure supplement 1D*). IAPs contain a RING domain with E3 ubiquitin ligase activity which mediates their own K48 polyubiquitination and that of other protein targets and thus it is crucial for the role of IAPs in suppressing apoptosis (*Estornes and Bertrand, 2015*). To identify a possible post-transcriptional regulation of IAPs, we first compared the total amount of K48-ubiquitin-linked proteins in control and CM cells. Constricted cells displayed an accumulation of ubiquitinated proteins, indicating a possible bottleneck for protein degradation in CM-stressed cells (*Figure 2—figure supplement 1E*). In addition, we performed a chase assay with the protein synthesis inhibitor cycloheximide, in order to assess protein half-life (*Figure 2—figure supplement 1F*), using MCL-1 as a positive control as it is rapidly degraded by the proteasome.

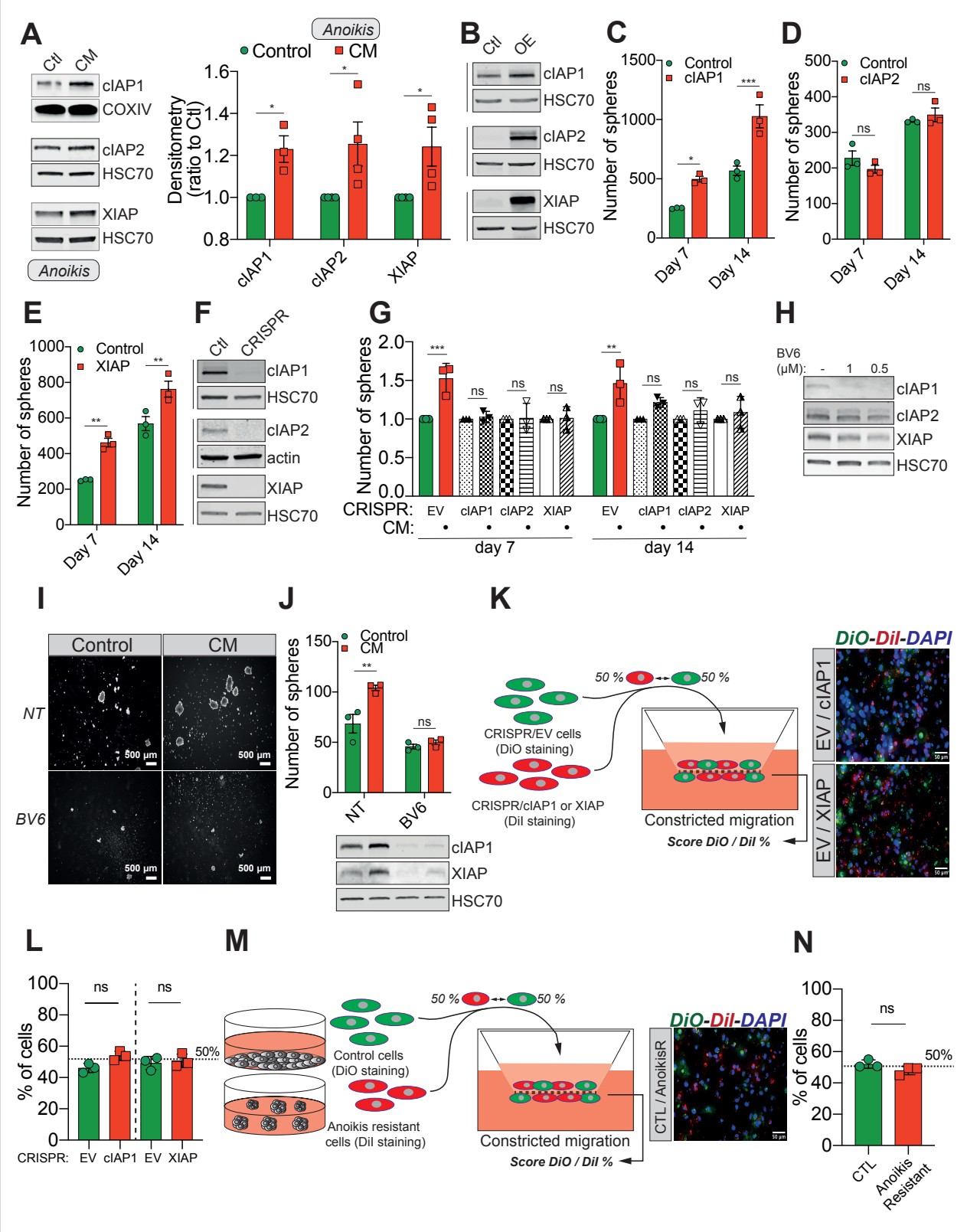

**Figure 2.** Confined migration (CM)-induced resistance to anoikis relies on Inhibitory of Apoptosis Proteins (IAPs). (**A**) Western blot analysis of IAPs protein expression in MDA-MB-231 cells after CM through 3 µm in diameter membranes, with cells grown in ultra-low attachment conditions (left panel) and the corresponding densitometry analysis (right panel) (*n* = 3–4, two-way analysis of variance [ANOVA] statistical test). (**B**) Validation of IAPs protein overexpression in MDA-MB-231 cells by Western blot. Quantitative comparison of resistance to anoikis in cIAP1 (**C**), cIAP2 (**D**), and XIAP (**E**)

*Figure 2 continued on next page*

*Figure 2 continued*

overexpressing MDA-MB-231 cells in steady-state conditions. Clonogenic structures were counted after 7 and 14 days of culture in ultra-low attachment condition (n = 3, two-way ANOVA statistical test). (**F**) Western blot analysis of protein expression validating the efficacy of CRISPR/Cas9-mediated deletion of IAPs. (**G**) Quantitative comparison of resistance to anoikis in MDA-MB-231 cells deleted for cIAP1, cIAP2, and XIAP using CRISPR/Cas9 and subjected to CM (n = 3, two-way ANOVA statistical test). (**H**) Validation by Western blot of IAPs expression inhibition by treating MDA-MB-231 cells with BV6. Two concentrations of BV6 (0.5 and 1 µM) were used for 16 hr of treatment. (**I**) Representative images of clonogenic structures from MDA-MB-231 control and CM cells cultured in low attachment conditions and treated with 0.5 µM of BV6. (**J**) Quantitative comparison of resistance to anoikis in CM MDA-MB-231 cells treated with 0.5 µM of SMAC mimetic BV6. Western blot analysis confirming inhibition of cIAP1 and XIAP expression following BV6 treatment (n = 3, two-way ANOVA statistical test). (**K**) CRISPR EV (control) and CRISPR cIAP1 or XIAP MDA-MB-231 cells were stained with DiO and DiI, respectively, mixed at a 50:50% ratio and then tested in CM. The cells undergoing CM were then scored for the ratio DiO:DiI and representative fluorescence images are shown. (**L**) Quantification of the percentage of CRISPR EV versus CRISPR cIAP1 and XIAP cells passing through constrictive pores (n = 3, t-test). (**M**) Control and anoikis-resistant cells (selected for 7 days in ultra-low attachment culture plates) were stained with DiO and DiI, respectively, mixed at a 50:50% ratio and then tested in CM. The cells undergoing CM were then scored for the ratio DiO:DiI and representative fluorescence images are shown. (**N**) Quantification of the percentage of control versus anoikis-resistant cells passing through constrictive pores (n = 3, t-test). (Statistical significance: ns - P > 0.05; * - P ≤ 0.05; ** - P ≤ 0.01; *** - P ≤ 0.001; **** - P ≤ 0.0001).

The online version of this article includes the following figure supplement(s) for figure 2:

**Figure supplement 1.** Confined migration (CM)-induced resistance to anoikis relies on Inhibitory of Apoptosis Proteins (IAPs).

When focusing on XIAP, which was the most differentially expressed IAP in CM-challenged cells, we observed a slower decrease in XIAP protein in CM cells compared to control cells, suggesting a lower proteasomal degradation (*Figure 2—figure supplement 1F*). XIAP may thus be more stable in CM cells, which might explain its higher expression following CM. CM-triggered resistance to anoikis therefore involves the pro-survival IAP proteins, which can be efficiently targeted by pre-clinically validated SMAC mimetics.

IAPs are commonly described to modulate NF-κB pathway, while in a positive feedback loop NF-κB regulates IAPs expression (*Diessenbacher et al., 2008*; *Gyrd-Hansen and Meier, 2010*). To test whether CM activates NF-κB, we first assessed p65 translocation from the cytoplasm to the nucleus upon CM, and showed an increase in cells displaying nuclear p65 (*Figure 2—figure supplement 1G*). Using two luciferase-based NF-κB reporter constructs, CM was also found to increase NF-κB transcriptional activity (*Figure 2—figure supplement 1H, I*). To test whether NF-κB activation was required for CM-driven resistance to anoikis, we expressed in MDA-MB-231 cells the IκB super repressor (IκB$^{SR}$), which is a non-degradable IκBα that blocks the nuclear shuttling of p65 (*Van Antwerp et al., 1996*). Accordingly, the stable expression of IκB$^{SR}$ blocked p65 nuclear shuttling following TNFα treatment (*Figure 2—figure supplement 1J*). However, in these settings, blocking the NF-κB pathway in CM cells did not prevent their resistance to anoikis (*Figure 2—figure supplement 1K,L*). In addition, the artificial activation of NF-κB pathway using TNFα treatment, can upregulate cIAP2 as previously described, yet it had no effect on both cIAP1 and XIAP (*Figure 2—figure supplement 1M, N*; *Diessenbacher et al., 2008*). These data suggest that CM-driven mechanical stress is characterized by NF-κB activation, which is not involved in the survival advantage observed in CM-challenged cancer cells.

An important issue is whether CM selects for a cellular population exhibiting higher IAPs expression and/or anoikis resistance or it specifically induces de novo these changes. If the observed phenotypes were the effect of a selection mechanism, this would imply that cells with higher IAPs expression or being more resistant to anoikis (these cells being pre-existent in a heterogeneous cell population) would have a migration advantage when undergoing CM. To test this, first we used the CRISPR/Cas9 cells described previously (*Figure 2F*), with CRISPR EV considered as high cIAP1 and XIAP-expressing cells. We stained the EV and the CRISPR cIAP1/XIAP with the lipophilic cyanide dyes DiO (green) and DiI (red), respectively, and then mixed them a known ratio of 50:50. The cellular mix was then subjected to CM and at the end the green-to-red ratio was reassessed (*Figure 2K*). As shown in *Figure 2L*, the score was similar to the initial 50:50 ratio, with IAPs high expressing cells (the EV cells) showing the same CM capacity as IAPs KO cells. In a complementary approach, to determine whether CM selects cancer cells initially displaying an increased resistance to anoikis, we enriched in anoikis-resistant cells by continuously culturing MDA-MB-231 cells in ultra-low attachment plates for 7 days (*Figure 2M*). As described above, control adherent and anoikis-resistant cells were stained with DiO and DiI, respectively, mixed in a 50:50 ratio and tested in CM (*Figure 2M*). This time as well we did not observe any CM advantage driven by resistance to anoikis, as the green-to-red ratio of cells

undergoing CM is identical to the initial one (*Figure 2N*). These results imply that CM is not selecting for cells having a different expression of IAPs or displaying resistance to anoikis.

To conclude, resistance to anoikis driven by CM mechanical stress relies on the pro-survival function of IAPs, regulated at the post-transcriptional level following mechanical stress.

## CM enhances the aggressiveness of breast cancer cells and promotes evasion from immune surveillance

To gain mechanistic insights into the relationship between cellular constriction and resistance to anoikis, we performed RNA sequencing (RNA seq) analysis on MDA-MB-231 cells undergoing CM, compared to control cells. Remarkably, CM cells displayed an almost global inhibition of transcription, making their transcriptional profile distinct from control cells (*Figure 3—figure supplement 1A, B*). To investigate this effect further, we performed a Western blot analysis for histone H3 epigenetic modifications, associated with transcriptional activation (H3K27 acetylation) or heterochromatin (H3K9 trimethylation). Consistently with their overall transcriptional inhibition, CM cells had a lower histone H3 acetylation and a reduction in heterochromatin (lower H3K9 me3), which may indicate a decrease in nuclear stiffness, which is needed when cells navigate through narrow spaces (*Figure 3—figure supplement 1C*). When querying the Gene Ontology (GO) Biological Processes using both Enrichr (*Figure 3—figure supplement 1D*) and g:Profiler (*Figure 3—figure supplement 1E*), several pathways associated with cellular motility such as 'Extracellular matrix organization', 'Cell-matrix adhesion', or 'Cell adhesion' were significantly overrepresented in CM cells (*Chen et al., 2013*; *Raudvere et al., 2019*). Given that metastatic cells acquire an aggressive phenotype, we thus hypothesized that CM may impact cancer cell motility (*Lambert et al., 2017*).

Since external mechanical forces are responsible for rapid cytoskeleton rearrangement, we first stained F-actin in control and CM-stressed MDA-MB-231 cells (*Torrino et al., 2021*). This revealed an increased number of stress fibres following CM, in addition to more abundant filopodia (*Figure 3A*). By using Nanolive imager-based cellular tomography, we uncovered that cancer cells subjected to CM had a higher variation in cell area over time (*Figure 3B*). To test whether these phenotypic changes were accompanied by increased cell motility, single-cell migration was first tracked in control and CM cells over a 24-hr period. CM-challenged cancer cells displayed a significantly higher velocity and travelled further than control cells (*Figure 3C–E*). Proliferating cancer cells adhere to their substrate via focal adhesions that are equally important during migration, especially metastasis (*Devreotes and Horwitz, 2015*; *Roussos et al., 2011*). Here, we used immunofluorescence for two key components of focal adhesions, namely paxillin and vinculin (*Figure 3—figure supplement 1F*), to assess the number of focal adhesions following CM. The increased single-cell migration observed was not correlated with the number of focal adhesions. In contrast to single-cell migration, breast cancer cells subjected to a single round of CM did not outperform control cells when assessed by collective cell migration and invasion (*Figure 3F–K*). As cells encounter several mechanical challenges during metastasis, we then imposed three consecutive CM passages on MDA-MB-231 cells, and found that challenged cells had a significant gain in chemotaxis and collective cell migration (*Figure 3L–N*, experimental setup in *Figure 3—figure supplement 1G*). In an effort to determine other features that might facilitate CM, we focused on nuclear lamins, which are important regulators of nuclear stiffness and shape during CM (*Harada et al., 2014*; *de Leeuw et al., 2018*). To establish whether lamins have a role during MDA-MB-231 CM, we stably overexpressed lamin A GFP (*Figure 3—figure supplement 1H*) and set up a competition-like scenario between lamin A GFP overexpressing and control cells stained in red with DiI, similar to the experiments performed in *Figure 2K, M* (*Figure 3—figure supplement 1I*). Confirming previous studies, we also found that MDA-MB-231 cells with higher levels of lamin A have a lower capacity to perform CM, most probably due to increased nuclear stiffness (*Figure 3—figure supplement 1J*). Yet, when the expression of lamin A/C was tested in cancer cells right after CM, it was similar to that found in control cells, implying once more that CM does not select for cells with certain characteristics, such as lower expression of lamins (*Figure 3—figure supplement 1K*).

Two of the gene expression signatures over-represented in breast cancer cells undergoing CM involve the regulation of T-cell-mediated immunity and most importantly the negative regulation of NK cells mediated cytotoxicity (*Figure 3—figure supplement 1D, E*). The immune system plays a crucial role in preventing metastatic dissemination through a process called immune surveillance. The innate immune NK cells and the adaptive ones, T cells αβ (CD4+ and CD8+), as well as γδ T

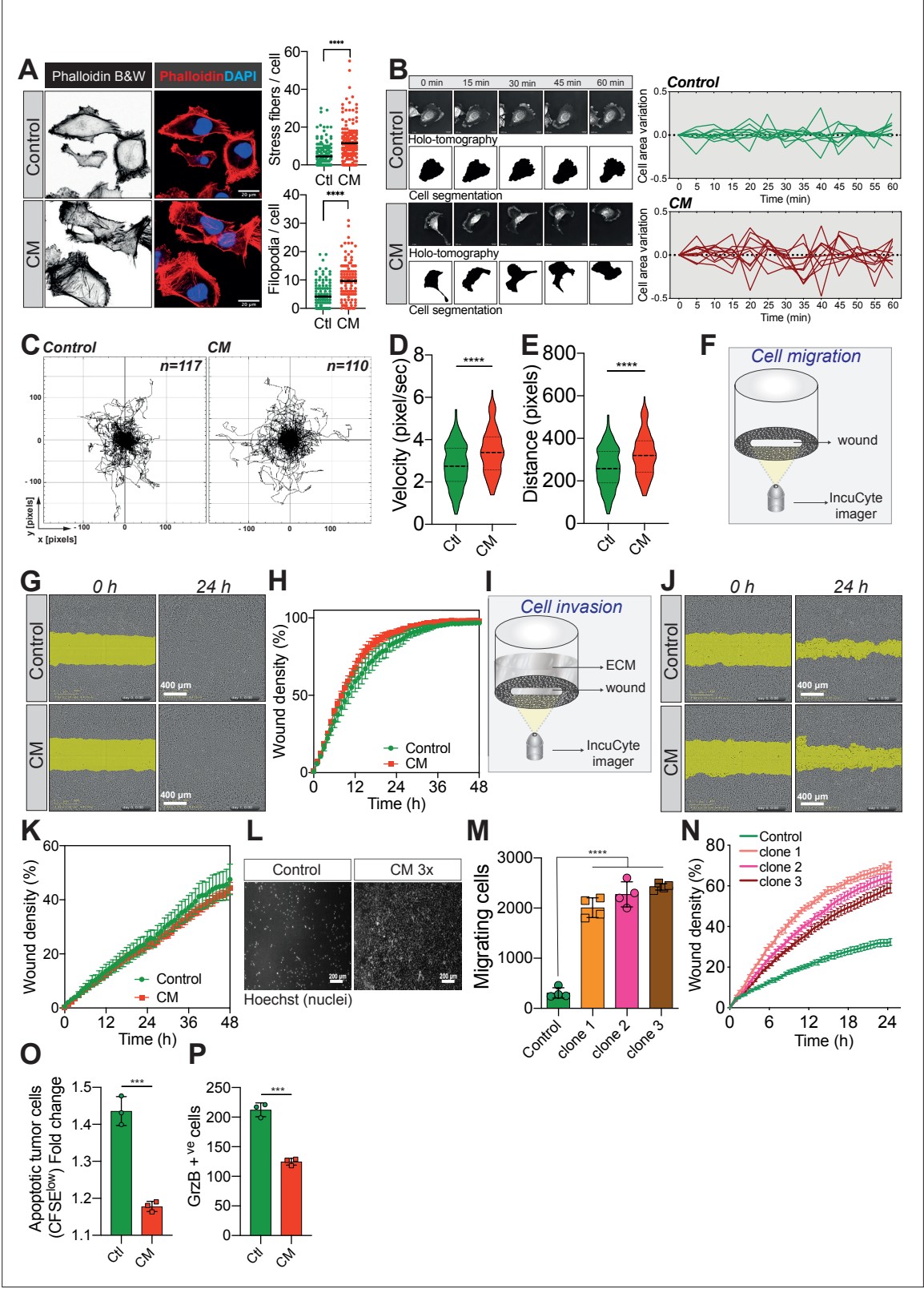

**Figure 3.** Confined migration (CM) confers breast cancer cells with a discrete aggressive behaviour. (**A**) Representative immunofluorescence images of MDA-MB-231 cells after CM with phalloidin-stained actin filaments. The right panels are quantification of the number of stress fibres and filopodia. (**B**) Representative kinetics holotomographic images (phase and cell segmentation) of control and MDA-MB-231 cells subjected to CM obtained by Nanolive imaging (left panel). Corresponding quantitative analysis of cell area variations of control and CM cells during a 60-min time-lapse acquisition

*Figure 3 continued on next page*

*Figure 3 continued*

is shown in the right panel. (**C**) Spider plot analysis for single-cell migration assay. Control (117 cells) and CM (110 cells) cells were tracked for 24 hr. Quantitative analysis of single-cell migration velocity (**D**) and distance travelled (**E**) between control and CM MDA-MB-231 cells. (**F**) Schematic representation of the IncuCyte ZOOM imager-based wound-healing assay. (**G**) Monolayers of MDA-MB-231 control and CM cells were wounded and pictures were taken immediately after wound induction (T0) and 24 hr later. (**H**) Corresponding quantitative analysis of the migratory potential of control and CM cells through wound area measurement (*n* = 3, a representative experiment is shown). (**I**) Schematic representation of invasion assay. After the wound was made, breast cancer MDA-MB-231 cells invaded through a matrigel plug until they closed the wound. (**J**) Monolayers of MDA-MB-231 control and CM cells were wounded and pictures were taken immediately after wound induction (T0) and 24 hr later. (**K**) Corresponding quantitative analysis of the invasive potential of control and CM cells through wound area measurement (*n* = 3, a representative experiment is shown). (**L**) Representative images of control and MDA-MB-231 cells subjected to serial CM and undergoing chemotaxis through transwell membranes with 8 µm in diameter pores. (**M**) Chemotaxis quantification relative to (**I**) (*n* = 3, one-way analysis of variance [ANOVA] statistical test). (**N**) Comparative quantitative analysis based on IncuCyte ZOOM imager of the migratory potential between control and MDA-MB-231 cells that have undergone serial CM (for three different clones) (*n* = 3, a representative experiment is shown). (**O**) FACS analysis of apoptotic cells among CM tumour cells compared to control cells, co-cultured with natural killer (NK cells at the ratio of 1:20 for NK cells). Results were analysed by assessing the ratio of CFSE$^{low}$/CFSE$^{high}$ with baseline-correction to no NK cell culture condition (*n* = 3, *t*-test). (**P**) FACS analysis of GrzB+ tumour cells among CM and control MDA-MB-231 cells, co-cultured with NK cells at a ratio of 1:20 (*n* = 3, *t*-test). (Statistical significance: ns - P > 0.05; * - P ≤ 0.05; ** - P ≤ 0.01; *** - P ≤ 0.001; **** - P ≤ 0.0001).

The online version of this article includes the following figure supplement(s) for figure 3:

**Figure supplement 1.** Confined migration (CM) confers breast cancer cells with a discrete aggressive behaviour.

lymphocytes are the unique actors of this phenomenon (*Pagès et al., 2005*; *Barrow et al., 2018*). The advantage of NK-mediated immune surveillance is that it is very effective on cancer cells that are in the blood circulation (*Garrido and Aptsiauri, 2019*). We therefore reasoned that CM might also influence NK-mediated immune surveillance. To test this, we co-cultured control and CM-challenged cells stained with carboxyfluorescein succinimidyl ester (CFSE) with primary NK cells, obtained from healthy donor blood. Interestingly, breast cancer cells were partially protected from the NK-mediated cytotoxicity following CM (*Figure 3O*, CFSE$^{low}$ population represents apoptotic cells), consistent with lower levels of toxic granzyme B incorporation (*Figure 3P*).

Taken together, these data demonstrate that a single event of CM has a profound effect on single-cell migration, while several rounds of CM enhance cancer cell chemotaxis and collective migration. In addition, CM contributes to evasion from NK-mediated immune surveillance.

## Breast cancer cells subjected to CM have an increased metastatic potential in vivo

We next wondered whether the effect of CM on the in vitro breast cancer aggressiveness was applicable in vivo. We injected control and MDA-MB-231 cells subjected to one round of CM into the tail vein of immune-deficient mice and then analysed lung metastatic colonization by micro-computed tomography (microCT). We observed that metastasis incidence was significantly higher for CM cells 6 weeks post-engraftment (*Figure 4A*). Moreover, the volume of healthy lung tissue in mice engrafted with CM breast cancer cells was considerably smaller than control counterparts, indicating their increased aggressiveness (*Figure 4B, C*). Of note, this was also the case for breast cancer cells experiencing three consecutive rounds of CM (*Figure 4—figure supplement 1A*). In addition, the increased aggressiveness of constricted cells was also quantified by measuring the area of metastatic lesions following H&E staining (*Figure 4D, E*).

Collectively, these data demonstrate that a single event of CM was sufficient to enhance lung metastatic colonization in mice. Hence, we show here that CM is characterized by resistance to anoikis, increased single-cell motility, NF-κB activation and escape from immune surveillance. Although independent of NF-κB activation, the resistance to anoikis relies on pro-survival IAPs, regulated at the post-transcriptional level following mechanical stress. Overall, these events contribute to enhancing breast cancer cell aggressiveness (*Figure 4F*).

## Discussion

The role of cell death in cancer has been extensively investigated and its inhibition by cell-autonomous mechanisms such as overexpressing IAPs or anti-apoptotic BCL2 family proteins is a stepping stone for oncogenesis (*Gyrd-Hansen and Meier, 2010*; *Strasser and Vaux, 2020*). However, very little

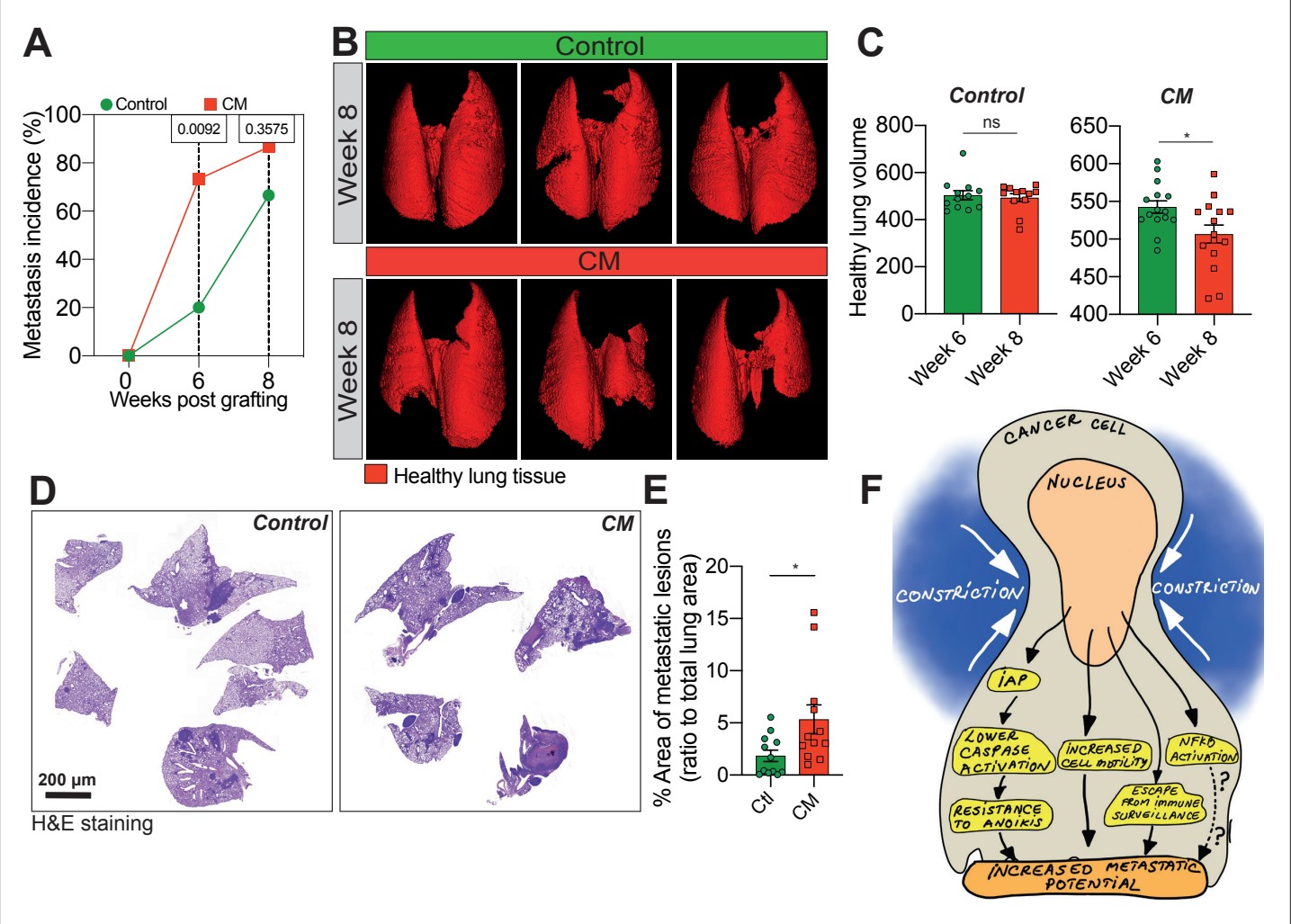

**Figure 4.** Breast cancer cells subjected to confined migration (CM) acquire enhanced metastatic potential. (**A**) Analysis of lung metastasis incidence in nude mice engrafted with either control or CM MDA-MB-231 cells (two-tailed Fisher's exact test). (**B**) Representative micro-computed tomography (microCT)-based 3D reconstructions (red represents healthy lung volume) of lungs from mice engrafted with either control or MDA-MB-231 cells undergoing CM, 15 mice/condition. (**C**) Corresponding quantification of remaining healthy lung volume 6 and 8 weeks post-engraftment. (**D**) Representative H&E staining of lung sections. (**E**) Quantification of lung metastatic foci (ratio to total lung surface). (**F**) Model: as a consequence of CM but not compression, cancer cells become resistant to cell death triggered by loss of cell attachment (anoikis), which relies on increased expression of IAPs proteins. NF-$\kappa$B is also activated by mechanical stress, yet it does not impact resistance to anoikis. In addition, CM cancer cells are more resistant to natural killer (NK)-mediated immune surveillance. Together with a marked motility advantage, this confers an increased metastatic colonization advantage to breast cancer cells having undergone CM. (Statistical significance: ns - P > 0.05; * - P ≤ 0.05; ** - P ≤ 0.01; *** - P ≤ 0.001; **** - P ≤ 0.0001).

The online version of this article includes the following figure supplement(s) for figure 4:

**Figure supplement 1.** Breast cancer cells subjected to confined migration (CM) acquire enhanced metastatic potential.

is known on how the mechanically challenging tumour microenvironment impacts efficient lethal caspase activation and cancer cell death, and how this might favour cancer aggressiveness. This is a timely issue since the causal link between tumour stiffening, mechanical stress, and cancer progression is well documented (*Paszek et al., 2005*; *Acerbi et al., 2015*). As the tumour stiffness and the inherent mechanical stress recently gained notoriety in favouring cancer progression, one could wonder whether this pro-oncogenic effect may be partly attributed to an underappreciated inhibitory effect on efficient induction of tumour cell death (*Shen and Kang, 2020*; *Gensbittel et al., 2021*; *Nia et al., 2020*).

Here, we characterized the impact of mechanical stress, mimicking that encountered either within the primary tumour (compression) or during metastasis (CM), on the acquisition of tumourigenic

properties, in particular resistance to cell death. Our experimental setup, based on commercially available transwell membranes with pores of 3 µm in diameter, forced breast cancer cells to undergo severe CM towards a chemotactic cue. Although it was previously reported that CM caused nuclear lamina breaks and widespread DNA damage, the MDA-MB-231 cells subjected to CM used herein recovered well from the forced passage, lacked obvious apoptotic caspase activation, and displayed no major differences in cell proliferation (*Raab et al., 2016*; *Denais et al., 2016*; *Harada et al., 2014*; *Pfeifer et al., 2018*).

In addition to the CM that cells undergo to exit the primary tumour, CTCs also need to survive the complete loss of cell attachment, which normally triggers anoikis. Since resistance to anoikis was previously described to protect CTCs and favour their metastatic seeding, we tested this in breast cancer cells challenged by CM and unveiled a significant resistance to anoikis (*Paoli et al., 2013*). As anoikis is a variation of apoptosis that relies on lethal caspase activation, we found that CM-driven resistance to anoikis was also mirrored by an inhibition of caspase activation. Although it has been reported that resistance to anoikis could be promoted by shear stress, we report here for the first time that CM, a mechanical stress in the metastatic cascade happening before the fluid shear stress, can also induce resistance to anoikis in breast cancer cells (*Li et al., 2019*). Conversely, cancer cells migrating through larger 8 µm in diameter microporous transwells, which do not induce CM, did not acquire resistance to anoikis. Aside from CM, cancer cells are subjected in situ to an important compressive stress within a rapidly growing tumour (*Nia et al., 2020*). Regarding cell death sensitivity, however, compression did not impact the resistance to anoikis, thus discriminating both types of mechanical stress. Interestingly, resistance to anoikis was transient since it remained significant up to 3 days following the mechanical challenge. This reversibility suggests that a temporary epigenetic, transcriptional, and/or translational programme is induced following acute constriction.

We then sought to uncover the cell-autonomous pro-survival pathways engaged in CM-challenged breast cancer cells and we narrowed them down to the pro-survival IAPs. Interestingly, these proteins were described to promote resistance to anoikis in several cancers. Nevertheless this is the first study showing that mechanical stress increases IAPs expression, especially cIAP1 and XIAP, likely by modifying their protein turnover (*Toruner et al., 2006*; *Liu et al., 2006*; *Berezovskaya et al., 2005*). In an effort to revert resistance to anoikis and restore caspase-dependent cell death, we used SMAC mimetics developed to efficiently deplete IAPs. Currently, several SMAC mimetics such as birinapant and LCL161 are in phase two clinical trials for ovarian cancer and myeloma (source: https://clinical-trials.gov/). Importantly, we found that BV6 treatment re-sensitized cancer cells subjected to CM to anoikis, reinforcing the relevance of SMAC mimetics for clinical use. Moreover, we found that breast cancer cells experiencing CM activate the NF-κB pathway, which is likely due to DNA damage (*Pfeifer et al., 2018*; *Hadian and Krappmann, 2011*). However, our results indicate that NF-κB activation is dispensable for the acquired resistance to anoikis, yet we cannot exclude additional pro-survival effects.

In this study, we also made an attempt to establish whether CM is actively inducing changes in IAPs protein levels and resistance to anoikis or simply it selects and enriches in a pre-existing cellular population displaying these characteristics. The experiments shown in *Figure 2K–N* and *Figure 3— figure supplement 1I–K* do not support the scenario of a selection process, yet we cannot completely exclude that CM selects for cells having other characteristics indirectly influencing IAPs expression or anoikis resistance.

To gain a clearer insight into CM-associated gene regulation, control and cancer cells experiencing CM were analysed by RNA sequencing. CM had a dramatic effect of overall transcriptional inhibition, which is most probably the immediate effect of DNA damage and altered chromatin organization (*Shah, 2021*; *Hsia et al., 2021*; *Heine et al., 2008*). Indeed, this was mirrored by the marked reduction of acetylated histone H3, illustrating global transcriptional inhibition. However, the duration of transcriptional inhibition following acute cell constriction remains to be investigated. In addition, it would be relevant to test by ChIP-Seq the exact gene regulatory networks impacted by loss of histone acetylation at promoter regions.

Since several gene expression signatures in CM cells were focused on cell adhesion and extracellular matrix disassembly, we next established whether breast cancer cells acquired migratory and invasive properties after recovering from a single round of CM. In line with published data, these cells displayed modifications of their cellular motility, with most obvious differences observed for F-actin

filament remodelling and single-cell migration (*Rudzka et al., 2019*; *Liu et al., 2015*; *Tse et al., 2012*). When CM was applied three consecutive times, the recovered cells were very aggressive, possibly due to the acquisition of a more stable aggressive transcriptional and/or epigenetic programme. This illustrates that multiple CM events encountered by a cancer cell during its dissemination to distant organs can have dramatic effects on its aggressive phenotype. These data support the existing relationship between mechanical stress and increased aggressiveness (*Nader et al., 2021*; *Tse et al., 2012*). A recent study from Nader et al. elegantly showed that invasive foci in breast cancer are enriched in constricted cells characterized by deformed nuclei and frequent TREX1-dependant DNA lesions. Relevant to our study, these cells were in a partial EMT state that enhanced their invasion potential (*Nader et al., 2021*).

Aside from the capacity to overcome anoikis and discrete modifications in motility, cancer cells experiencing a single CM event were also protected from NK-mediated immune surveillance. Despite these promising results, further research should be undertaken to evaluate NK function and analyse NK cytokine secretion in the presence of mechanically stressed cells. Since NK immune surveillance is conditioned by the expression on the cancer cell surface of MHC/HLA class I molecules and activation ligands, their expression should also be profiled following a mechanical stress (*Garrido and Aptsiauri, 2019*).

Finally, we tested whether CM-triggered resistance to anoikis, increased single-cell motility and evasion from NK-mediated immune surveillance were reflected in vivo by increasing the metastatic potential of invading cells. This was indeed the case since MDA-MB-231 breast cancer cells subjected to a single round of CM had a significant advantage to form metastatic lesions in the lungs of nude mice. This was also evidenced for cancer cells experiencing several rounds of CM.

In summary, this study refines our understanding of the pathophysiological relationship between mechanical stress and cancer aggressiveness. Our model of mechanical stress mimicking CM had an unexpected effect on resistance to anoikis, which was then mirrored by an enhanced metastatic seeding. In addition, our findings unveiled a previously unknown reliance of mechanically challenged breast cancer cells on IAPs for survival that could be targeted by treatment with SMAC mimetics.

## Materials and methods
### Cell lines
Human breast cancer cells MDA-MB-231 and Hs 578T (both a gift from P. Mehlen, CRCL, Lyon) were maintained in RPMI supplemented with 2 mM L-glutamine (Thermo Fisher Scientific, 25030-24), non-essential amino acids (Thermo Fisher Scientific, 11140-035), 1 mM sodium pyruvate (Thermo Fisher Scientific, 11360-039), 10% fetal bovine serum (FBS) (Eurobio, CVFSVF00-01), and 1% penicillin/streptomycin (Thermo Fisher Scientific, 15140-122).

### Stable cell line generation by lentiviral transduction
293T cells ($1.5 \times 10^6$ in a 10-cm Petri dish) were transfected with lentiviral plasmids together with pVSVg (Addgene, 8454) and psPAX2 (Addgene, 12260) using Lipofectamine 2000 (Thermo Fisher Scientific, 11668019) according to the manufacturer's instructions. Twenty-four and 48 hr later, virus-containing supernatant was collected, filtered, supplemented with 1 µg/ml polybrene (Sigma-Aldrich, H9268), and used to infect target cells. Two days later, the transduced cells were selected by growth in the appropriate antibiotic.

### Plasmid transfection
For the transient overexpression of cIAP1 and XIAP, MDA-MB-231 cells ($1.2 \times 10^6$) were plated overnight on a 10-cm Petri dish. The cells were then transfected using Lipofectamine 2000 (Thermo Fisher Scientific, 11668019) with pcDNA3 as empty vector or PEF-hXIAP-Flag for XIAP. A co-transfection with PV2L-Blasti-TRAF2 and pEF6 2xHA cIAP1 WT was needed for overexpressing cIAP1. Six hours later, the transfection medium was replaced by fresh medium and the cells were allowed to grow for 48 h.

### Generation of CRISPR/Cas9-based KO cells
The oligos containing the gene-specific sgRNA target were cloned into the LentiCRISPRv2 Blasticidin (Addgene, 83480) as previously described (*Shalem et al., 2014*). Following lentiviral transduction,

**Table 1.** List of CRISPR primers.

| Gene of interest | Forward primer (5′–3′) | Reverse primer (5′–3′) |
| --- | --- | --- |
| BAX | CACCGAGTAGAAAAGGGCGACAACC | AAACGGTTGTCGCCCTTTTCTACTC |
| BAK1 | CACCGGCCATGCTGGTAGACGTGTA | AAACTACACGTCTACCAGCATGGCC |
| BIRC2 | CACCGCATGGGTAGAACATGCCAAG | AAACCTTGGCATGTTCTACCCATGC |
| BIRC3 | CACCGCATGGGTTCAACATGCCAAG | AAACCTTGGCATGTTGAACCCATGC |
| XIAP | CACCGTATCAGACACCATATACCCG | AAACCGGGTATATGGTGTCTGATAC |

cells were selected with 10 mg/ml blasticidin (Invivogen, ant-bl) for 2 weeks prior to analysis. The CRISPR/Cas9 primers are presented in *Table 1*.

## Transwell assays

Breast cancer cells (MDA-MB-231, Hs578T; $3 \times 10^6$) were plated on a 75-mm transwell insert with a polycarbonate membrane pore size of 3 µm (Corning, 3420). Before seeding, the insert was coated with a layer of matrigel (300 µg/ml). A gradient of serum was then created between the two compartments of the transwell (0% FBS in the top compartment and 20% below) and renewed 5 hr later. After 72 hr, cells were harvested after washing the insert with phosphate-buffered saline (PBS) and incubation with trypsin. Cells that have migrated through the 3 µm pores were designated as the 'constricted cells', whereas the ones that did not migrate were the 'control cells'.

For the 8-µm transwell experiments, $5 \times 10^5$ MDA-MB-231 cells were plated on a 6-well insert with a polycarbonate membrane pore size of 8 µm (Greiner Bio One 657628) and the cells were recovered and analysed 48 hr later.

For certain experiments, cells recovered for the transwell assay were stained with Hoechst 33,342 (10 µg/ml, Thermo Fisher Scientific, H1399), Calcein AM (0.4 mg/ml, Life, C1430), and Vybrant cell-labelling solutions (DiI and DiO, V-22885 and V-22886) according to the manufacturer's instruction.

## Anoikis assay

MDA-MB-231 and Hs578T cells ($2 \times 10^5$) were seeded onto a 6-well Clear Flat Bottom Ultra-Low Attachment plate (Corning, 3471) in complete RPMI medium. Cells then formed 3D clonogenic structures that were imaged and scored after 7 and 14 days of culture. Six wells were plated for each condition and experiments were repeated three times for each cell line. Cells were also grown on a 1% agarose Petri dish in RPMI medium without serum for 24 hr and proteins were extracted with protein lysis buffer.

## Soft agar colony assay

Cells ($10^3$/well) were suspended in 1 ml of 0.3% low gelling temperature agarose (Sigma, A9414) and plated onto a 1% agarose layer in three wells of a 6-well plate. When the 0.3% agarose solidified, the wells were covered in complete RPMI media and colonies were scored 4 weeks later.

## Immunofluorescence

MDA-MB-231 ($5 \times 10^4$) were seeded onto coverslips placed in 24-well plate overnight. For studies on focal adhesions, the coverslips were first coated with 100 µg/ml matrigel (Sigma-Aldrich, E1270). After washing in PBS, cells were fixed in 4% PFA for 5 min and then washed once. Cells were permeabilized with 0.2% Triton X-100 (Pan Reac, A4975.01) diluted in PBS, for 10 min at room temperature and the blocking of non-specific binding sites was done using 2% bovine serum albumin (BSA) in PBS, for 1 hr at room temperature. Cells were then incubated with the primary antibody COX IV (Cell Signaling, 4850 S), cytochrome *c* (Cell Signaling, 12,963 S), Alexa Fluor 647 Phalloidin (Invitrogen, A22287), vinculin (Sigma-Aldrich, V9131), paxillin (BD Transduction Biosciences, 610052), or p65 at 1/400–500 dilution in PBS, for 1 hr at room temperature or overnight at 4°C. Next, the cells were washed in PBS three times and then incubated with the appropriate secondary antibody coupled to Alexa Fluor (1/300, Thermo Fisher scientific, A21151 and A31571) for 1 hr at room temperature protected from light. The staining of nuclei was done with Hoechst 33,342 (10 µg/ml, Thermo Fisher Scientific, H1399)

or with DAPI mounting medium (Vectashield). The coverslips were finally mounted using Fluoromount (Southern Biotech, 0100-01). Slides were left to dry overnight before image acquisition using a Zeiss Axio Imager microscope (Zeiss).

## Anoikis resistance assay using the IncuCyte ZOOM imager

MDA-MB-231 cells ($10^4$ cells) were plated in a 96-well Clear Round Bottom Ultra-Low Attachment Microplate (Corning, 7007). SytoxGreen (30 nM, Life, 1846592) was also added to the medium to stain apoptotic cells. Cells were then imaged every 60 min using the IncuCyte ZOOM imager.

## VC3AI reporter-based caspase activation assay

MDA-MB-231 VC3AI (control and constricted) cells ($2 \times 10^5$) were collected from transwell assay and the mean fluorescence intensity of the green signal (VC3AI) was then determined by FACS Calibur flow cytometry (BD Biosciences, San Jose, CA, USA). Control cells were treated with 1 µM Actinomycin D as a positive control for cell death.

## Fluorometric caspase-3/7 activity assay

The cell pellets were resuspended in Cell Lysis Buffer before evaluating the amount of protein in each sample. Twenty µg of proteins were then mixed with Reaction Buffer supplemented with DEVD-AFC substrate. After 1 hr of incubation at 37°C, the caspase-3/7 activity was assessed by fluorescence measurement. Caspase-3 activity was determined using the caspase-3/CPP32 Fluorometric assay kit according to the manufacturer's instructions (BioVision, K105).

## Mitochondrial membrane potential assay

MDA-MB-231 cells ($10^5$) were harvested and resuspended in 0.1 µM tetramethylrhodamine ethyl ester perchlorate (TMRE, Thermo Fisher Scientific, T669) for 30 min at 37°C. This fluorescent compound accumulates only in intact mitochondria and highlights the mitochondrial membrane potential of living cells. CCCP (carbonyl cyanide 3-chlorophenylhydrazone) was used as a mitochondrial membrane potential disruptor. When mitochondria are depolarized, leading to a decrease in membrane potential, TMRE accumulation is reduced. After washing, the membrane potential (mean fluorescence intensity of the red signal) was determined by flow cytometry.

## Evaluation of mitochondrial superoxide levels

MDA-MB-231 (control and constricted) cells ($5 \times 10^4$) were harvested and resuspended in 5 µM Mitosox (Thermo Fisher Scientific, M36008) for 15 min at 37°C. After washing with PBS, the level of mitochondrial superoxide was determined by flow cytometry. Cells treated 1 hr with 500 µM $H_2O_2$ were used as a positive control for ROS production.

## Measurement of total ROS in live cells using CellROX staining

After plating $5 \times 10^4$ MDA-MB-231 cells (control and constricted) overnight in a 12-well plate, cells were trypsinized and treated 30 min at 37°C with 5 µM CellROX Deep Red reagent (Life Technologies, C10422) diluted in medium. This cell-permeant dye exhibits a strong fluorescence once oxidized by cytosolic ROS. Positive control of ROS consisted in cells treated 1 hr with 500 µM $H_2O_2$ (Sigma, H1009) before staining. Cells were then centrifuged and washed in PBS three times, before being resuspended in 200 µl of medium. The subsequent analysis was performed using FACS Calibur.

## ATP assay

ATP levels were measured in control and constricted breast cancer cells ($3 \times 10^5$ cells) using the ATP fluorometric assay kit (Sigma-Aldrich, MAK190), following the manufacturer's instructions. Cells treated 1 hr with 500 µM $H_2O_2$ served as a negative control. To evaluate the ATP concentration, an ATP standard curve from 0 to 10 nM was used.

## Cell cycle analysis

$10^5$ MDA-MB-231 cells (control and constricted) were washed once in PBS and pelleted in FACS tubes. Cold ethanol at 100% was added drop by drop while vortexing at a final concentration of 70% in PBS. Cells were then stored at −20°C until use. For FACS analysis, the cells were centrifuged at 1500 rpm

to remove ethanol, rinsed with PBS and then centrifuged at 2200 rpm for 5 min. The pellet was then treated with 100 μg/ml ribonuclease A (A8950) in order to specifically stain DNA. Propidium iodide (Sigma, P4864) was added at 100 μg/ml and cells were immediately analysed by flow cytometry.

## Cell compression

MDA-MB-231 cells ($3 \times 10^6$) were plated on 75-mm transwell inserts with a polycarbonate membrane pore size of 3 μm (Corning, 3420) that allows media and gas exchange during compression. Twenty-four hours later, a 2% agarose (Sigma, A9539-100G) disk was placed on top of the cells in order to prevent the direct contact with the plastic cup (3D printed by F. B.) placed above the agarose disk. A range of pressure was then tested for 24 hr (0, 200, 300, 400, and 600 Pa) by adding the appropriate lead weights in the plastic cup. At the end of the compression time, the cells under the agarose disk were washed and collected for further analysis.

## Western blot analysis

Proteins were isolated by lysing cell pellets in RIPA lysis buffer (Cell Signaling, 9806 S) supplemented with phosphatase inhibitors complexes 2 and 3 (Sigma-Aldrich, P5726-1ML, P6044-1ML), dithiothreitol (DTT) 10 mM, and protease inhibitor cocktail (Sigma-Aldrich, 4693116001). The protein concentration was then determined using the Protein Assay dye Reagent Concentrate (BioRad, 50000006). Equal amounts (15–20 μg) of each sample were separated on 4–12% sodium dodecyl sulfate–polyacrylamide gels (BioRad) under denaturing conditions (SDS–PAGE sample loading buffer [VWR, GENO786-701] supplemented with 1 mM DTT). The gels were then transferred onto a nitrocellulose membrane using the Transblot Turbo Transfer System (BioRad, 1704150EDU). An incubation of 1 hr with Intercept blocking buffer (Licor, 927-70001) blocked non-specific binding sites before incubating the membranes with the primary antibody (1/1000 in Intercept T20 Antibody Diluent (Licor)) overnight at 4°C, under agitation. The primary antibodies used were: actin (Sigma-Aldrich, A3854), PARP-1 (Cell Signaling, 9532), caspase-3 (Cell Signaling, 9,62 S), GFP (Life, A11122), BAX (Cell Signaling, 2772 S), BAK (Cell Signaling, 12,105 S), HSP60 (Cell Signaling, 4870), K48-Ub (Cell Signaling, 8081 S), COX IV (Cell Signaling, 4850 S), cIAP1 (Cell Signaling, 7065T), cIAP2 (Cell Signaling, 3130T), XIAP (Cell Signaling, 14,334 S), HSC70 (Santa Cruz Biotechnology, sc-7298), H3K27ac (Diagenode, C15210016), H3K9me3 (Diagenode, C15200153), BCL-xL (Cell Signaling, 2764), BCL2 (Cell Signaling, 15071), MCL1 (Cell Signaling, 4572), and lamin A/C (Cell Signaling, 4777). The membranes were rinsed four times for 5 min in tris buffered saline, with Tween 20 (TBST) 0.1% and then incubated with appropriate secondary antibody coupled to IRDye 800CW or 680RD dye (Licor; 1/10,000) for 1 hr at room temperature under agitation and protected from light. Four extra washing steps in TBST 1% and one in TBS were performed before scanning the membrane by Odyssey Imaging System for near infrared detection.

## Cycloheximide chase assay

To determine protein half-life, control and constricted MDA-MB-231 cells ($5 \times 10^5$) were treated in ultra-low attachment conditions with cycloheximide (50 μg/ml) for different durations (0, 6, 16, 24, 33, and 48 hr) and protein extracts were analysed by Western blot.

## Dual luciferase reporter assay

MDA-MB-231 cells ($10^5$) were plated in 12-well plates for 24 hr and were then co-transfected with the NF-κB luciferase reporter-containing plasmid and a Renilla plasmid using Lipofectamine 2000. After 48 hr of transfection, the luciferase activity was assessed with the Dual luciferase reporter assay (Promega, E1910) following the manufacturer's instructions. Firefly luciferase activity was then normalized against Renilla luciferase activity.

## Holotomographic microscopy

MDA-MB-231 cells ($5 \times 10^4$) cells were seeded onto Fluorodishes (Ibidi GmbH, Gräfeling, Germany). Holotomographic microscopy was performed on the 3D Cell-Explorer Fluo (Nanolive, Ecublens, Switzerland) using a ×60 air objective at a wavelength of $\lambda = 520$ nm. Physiological conditions for live-cell imaging were maintained using a top-stage incubator (Oko-lab, Pozzuoli, Italy). A constant temperature of 37°C and an air humidity saturation as well as a level of 5% $CO_2$ were maintained throughout

imaging. Refractory index maps were generated every 5 min for 1 hr. Images were processed with the software STEVE.

## Single-cell migration assay

$10^3$ breast cancer cells were seeded onto a 96-well ImageLock plate (Sartorius, 4379) and imaged for 24 hr using the IncuCyte ZOOM-based time-lapse microscopy. The acquired time-lapse images were treated with a manual tracking plugin using the ImageJ software. About 100 cells/condition were followed for 30 min in order to determine the accumulated distance and their velocity.

## Wound-healing assay

MDA-MB 231 cells ($5.5 \times 10^4$) were seeded onto a 96-well imageLock plate (Sartorius, 4379) and grown for 24 hr until cell confluency was reached. A scratch was then performed in the cell monolayer using a WoundMaker (Sartorius, 4563), following the manufacturer's instructions. Wound closure was imaged and quantified using the IncuCyte ZOOM imaging system.

## Invasion assay

Wells of a 96-well imageLock plate (4379, Sartorius) were first coated with 100 µg/ml of Matrigel (Sigma-Aldrich, E609-10 ml). After 1 hr, MDA-MB-231 cells ($5.5 \times 10^4$) were seeded 24 hr prior to the assay. A wound was then performed in the cell monolayer with the WoundMaker and a new layer of Matrigel (800 µg/ml, 2.55 mm thickness) was deposited onto cells for 1 hr at 37°C to allow polymerization. The top of the cells was covered with complete medium and the invasion potential of cancer cells was evaluated and quantified using IncuCyte ZOOM-based time-lapse microscopy.

## RNA sequencing

RNA sequencing from control and mechanically challenged MDA-MB-231 cells was done by the CRCL Cancer Genomics core facility. The libraries were prepared from 600 ng total RNA using the TruSeq Stranded mRNA kit (Illumina) following the manufacturer's instructions. The different steps include the PolyA mRNA capture with oligo dT beads, cDNA double strand synthesis, adaptors ligation, library amplification, and sequencing. Sequencing was carried out with the NextSeq500 Illumina sequencer in 75-bp paired-end.

## Bioinformatics analysis

All genomic data were analysed with R/Bioconductor packages, R version 4.0.3 (2020-10-10) [https://cran.r-project.org/https://cran.r-project.org/; http://www.bioconductor.org/] on a linux platform (x86_64-pc-linux-gnu [64-bit]).

Illumina sequencing was performed on RNA extracted from triplicates of each condition. Standard Illumina bioinformatics analyses were used to generate fastq files, followed by quality assessment [MultiQC v1.7, https://multiqc.info/], trimming and demultiplexing. 'Rsubread' v2.4.3 was used for mapping to the hg38 genome and creating a matrix of RNA-Seq counts. Rsamtools v2.6.0 * was used to merge two bam files for each sample (run in two different lanes). Next, a DGElist object was created with the 'edgeR' package v3.32.1 [https://doi.org/10.1093/bioinformatics/btp616]. After normalization for composition bias, genewise exact tests were computed for differences in the means between groups, and differentially expressed genes were extracted based on a false discovery rate

**Table 2.** List of qRT-PCR primers.

| Gene of interest | Forward primer (5'–3') | Reverse primer (5'–3') |
|---|---|---|
| GAPDH | TGCACCACCAACTGCTTAGC | GGCATGGACTGTGGTCATGAG |
| ACTB | AGAGCTACGAGCTGCCTGAC | AGCACTGTGTTGGCGTACAG |
| HPRT | TGAGGATTTGGAAAGGGTGT | GAGCACACAGAGGGCTACAA |
| XIAP | TGAGGGAGACGAAGGGACTT | TTGTCCACCTTTTCGCGCC |
| BIRC2 | ATCGTGCGTCAGAGTGAGC | CTTCAGGGTTGTAAATCGCAGT |
| BIRC3 | CTCTGGGCAGCAGGTTTACAA | AGGTCTCCATTTTGAGATGTTTTGA |

(FDR)-adjusted p value <0.05 and a minimum absolute fold change of 2. All raw and processed RNA-Seq data have been deposited at the Gene Expression Omnibus (GEO) repository, under accession number GSE176081.

Rsamtools: Binary alignment (BAM), FASTA, variant call (BCF), and tabix file import. R package version 2.6.0. https://bioconductor.org/packages/Rsamtools.

### Quantitative RT-PCR

Total RNA extraction was performed using the Nucleospin RNA Macherey-Nagel kit (740955) and quantified by NanoDrop. The conversion of messenger RNA into cDNA was performed using the Sensifast cDNA synthesis kit (Bioline, BIO-65053). cDNA was then amplified by PCR using specific primers for each gene designed with Primer-blast software (https://www.ncbi.nlm.nih.gov/tools/primer-blast/) and listed in *Table 2*. GAPDH, ACTB, and HPRT were used as housekeeping genes. The thermal cycling steps included an initial polymerase activation step at 95°C for 2 min, followed by 40 cycles at 95°C, 5 s, and 60°C, 30 s. The qRT-PCR experiments were performed using SYBR Green and a Lightcycler96 (Roche, Indianapolis, USA).

### In vivo lung metastasis model and lung imaging

MDA-MB-231 control and constricted cells ($2.5 \times 10^4$) were suspended in 100 µl PBS and injected into the tail vein of NMRI nude female mice. To attain statistical significance of a p value between 0.05 and 0.02, we grafted 15 mice for each group. Sample-size calculation was performed as previously described (*Fitts, 2011*). The absence of mycoplasma in injected cells was controlled before injection in animals. Burden of lung metastasis was evaluated over time by X-ray microCT-Scan (Quantum FX, Perkin Elmer). Mice were anesthetized with a continuous flow of 2–4% isoflurane/air (1.5 l/min). The lungs were imaged in a longitudinal manner for 2 min with an exposure of 0.746 Gy and the obtained raw data were reconstructed with the following acquisition settings: a 24-mm FOV diameter, 512 slices, and 50 µm voxel. The resulting images were viewed and analysed using 'Analyze of Caliper' software (AnalyzeDirect) and the remaining healthy lung volume was quantified and 3D represented.

### Histological analyses

If limiting points were not observed, mice were euthanized 8 weeks post-engraftment. Lungs were fixed in 4% buffered formalin, paraffin embedded, and three 3-µm sections separated by 300 µm were stained with hematoxylin–eosin. The slides were scanned using the panoramic scan II (3D Histech). These were then analysed with CaseViewer 2.2.0.85100 software (3DHISTECH Ltd) for the detection of metastasis.

### NK-mediated immune surveillance

Control and constricted MDA-MB-231 cells were cocultured with human NK cells sorted from peripheral blood using the NK cell isolation Kit (Miltenyi Biotec 130-092-657) at the ratio of 1:20, in triplicate for each condition. Before co-culturing, tumour cells were pretreated with 10 µg/ml mitomycin for 1 hr to stop proliferation, and were incubated and tagged with CFSE (Invitrogen CellTrace, C34570) at 1 µl/ml for 20 min. Twenty-four hours later, cells in each condition were recovered by trypsin and stained intracellularly with Granzyme B (Biolegend, AF647, clone GB11). Flow cytometry was performed by BD LSR Fortessa HTS and data were analysed using GraphPad Prism V9.

### Image analysis

Image analysis was performed using the ImageJ software 1.52a.

### Statistical analysis

Data are expressed as the mean ± standard error of the mean. A two-tailed Student's *t*-test was applied to compare two groups of data. Analyses were performed using the Prism 5.0 software (GraphPad).

## Acknowledgements

This work was supported by funding from LabEx DEVweCAN (University of Lyon), Agence Nationale de la Recherche (ANR) Young Researchers Project (ANR-18-CE13-0005-01), La Ligue Nationale

Contre le Cancer, Fondation de France, and the National Cancer Institute (PLBIO21-003). We thank Brigitte Manship for reviewing the manuscript, Virgile Raufaste-Cazavieille, Léa Magadoux, Mathieu Deygas, and Thomas Barre (AniCan Image, Lyon, France, funding PHENOCAN ANR-11-EQPX-0035 PHENOCAN) and the Anatomopathology core facility for technical assistance.

## Additional information

### Funding

| Funder | Grant reference number | Author |
|---|---|---|
| Agence Nationale de la Recherche | LabEX DEVweCAN | Gabriel Ichim |
| Agence Nationale de la Recherche | ANR-18-CE13-0005-01 | Gabriel Ichim |
| Ligue Nationale Contre le Cancer | | Gabriel Ichim |
| Fondation de France | | Kevin Berthenet |
| French National Cancer Institute | PLBIO21-003 | Gabriel Ichim |
| French National Cancer Institute | PNP2019 | Ana Hennino |
| Fondation de France | | Ana Hennino |
| Centre Léon Bérard | C442 | Ana Hennino |
| China Scholarship Council | | Zhichong Wu |

The funders had no role in study design, data collection, and interpretation, or the decision to submit the work for publication.

### Author contributions

Deborah Fanfone, Data curation, Formal analysis, Funding acquisition, Investigation, Methodology, Writing - original draft, Writing – review and editing; Zhichong Wu, Investigation, Methodology, Writing – review and editing; Jade Mammi, Catherine Jamard, Investigation; Kevin Berthenet, Kathrin Weber, Andrea Halaburkova, Investigation, Writing – review and editing; David Neves, Investigation, Resources, Writing – review and editing; François Virard, Investigation, Methodology; Félix Bunel, Resources, Writing – review and editing; Hector Hernandez-Vargas, Conceptualization, Formal analysis, Investigation, Methodology, Software, Validation, Writing - original draft, Writing – review and editing; Stephen WG Tait, Resources; Ana Hennino, Conceptualization, Investigation, Methodology, Writing – review and editing; Gabriel Ichim, Conceptualization, Data curation, Formal analysis, Funding acquisition, Investigation, Methodology, Project administration, Resources, Supervision, Validation, Visualization, Writing - original draft, Writing – review and editing

### Author ORCIDs

Gabriel Ichim  http://orcid.org/0000-0002-8386-7405

### Ethics

All procedures were approved by the General Direction for Research and Innovation under the authorization number 24843-2020030913509726.

### Decision letter and Author response

Decision letter https://doi.org/10.7554/eLife.73150.sa1
Author response https://doi.org/10.7554/eLife.73150.sa2

## Additional files

### Supplementary files
- Transparent reporting form
- Source data 1. Uncropped western blot analysis.
- Source data 2. Raw numerical data.

### Data availability
Sequencing data have been deposited in GEO under accession codes GSE176081.

The following dataset was generated:

| Author(s) | Year | Dataset title | Dataset URL | Database and Identifier |
|---|---|---|---|---|
| Hernandez-Vargas H, Fanfone D, Ichim G | 2022 | Metastasis-relevant cellular constrictions as determinants for resistance to cell death and cancer aggressiveness | https://www.ncbi.nlm.nih.gov/geo/query/acc.cgi?acc=GSE176081 | NCBI Gene Expression Omnibus, GSE176081 |

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

# Appendix 1

**Appendix 1—key resources table**

| Reagent type (species) or resource | Designation | Source or reference | Identifiers | Additional information |
|---|---|---|---|---|
| Cell line (*Homo sapiens*) | MDA-MB-231 | ATCC and a gift from the laboratory of P. Mehlen (CRCL) | ATCC Cat# HTB-26, RRID:CVCL_0062 | |
| Cell line (*Homo sapiens*) | Hs578T | ATCC and a gift from the laboratory of P. Mehlen (CRCL) | ATCC Cat# HTB-126, RRID:CVCL_0332 | |
| Cell line (*Homo sapiens*) | 293T | Gift from the laboratory of Patrick Mehlen (CRCL, France) | ATCC Cat# CRL-3216, RRID:CVCL_0063 | |
| Recombinant DNA reagent | pCMV-VSV-G plasmid | pCMV-VSV-G was a gift from Bob Weinberg | RRID:Addgene_8454 | Envelope protein for producing lentiviral and MuLV retroviral particles |
| Recombinant DNA reagent | psPAX2 plasmid | psPAX2 was a gift from Didier Trono | RRID:Addgene_12260 | Second-generation lentiviral packaging plasmid |
| Transfected construct (human) | PEF-XIAP-Flag plasmid | Gift from John Silke | | |
| Transfected construct (human) | PV2L-Blasti-TRAF2 plasmid | Gift from Kevin Ryan | | |
| Transfected construct (human) | pEF6 2xHA cIAP1 plasmid | Gift from Pascal Meyer | | |
| Recombinant DNA reagent | LentiCRISPRv2 Blasticidin plasmid | LentiCRISPRv2 Blasticidin was a gift from Mohan Babu | RRID:Addgene_83480 | Mammalian expression of Cas9 and sgRNA scaffold |
| Sequence-based reagent | *BAX*_F | This article | CRISPR primer | CACCGAGTAGAAAAGGGCGACAACC |
| Sequence-based reagent | *BAX*_R | This article | CRISPR primer | AAACGGTTGTCGCCCTTTTCTACTC |
| Sequence-based reagent | *BAK1*_F | This article | CRISPR primer | CACCGGCCATGCTGGTAGACGTGTA |
| Sequence-based reagent | *BAK1*_R | This article | CRISPR primer | AAACTACACGTCTACCAGCATGGCC |
| Sequence-based reagent | *BIRC2*_F | This article | CRISPR primer | CACCGCATGGGTAGAACATGCCAAG |
| Sequence-based reagent | *BIRC2*_R | This article | CRISPR primer | AAACCTTGGCATGTTCTACCCATGC |
| Sequence-based reagent | *BIRC3*_F | This article | CRISPR primer | CACCGCATGGGTTCAACATGCCAAG |
| Sequence-based reagent | *BIRC3*_R | This article | CRISPR primer | AAACCTTGGCATGTTGAACCCATGC |
| Sequence-based reagent | *XIAP*_F | This article | CRISPR primer | CACCGTATCAGACACCATATACCCG |
| Sequence-based reagent | *XIAP*_R | This article | CRISPR primer | AAACCGGGTATATGGTGTCTGATAC |

*Appendix 1 Continued on next page*

*Appendix 1 Continued*

| Reagent type (species) or resource | Designation | Source or reference | Identifiers | Additional information |
|---|---|---|---|---|
| Other | Hoechst 33,342 | Thermo Fisher Scientific | H1399; CAS: 23491-45-4 | 10 µg/ml |
| Other | Calcein AM | Thermo Fisher Scientific | C1430 | 0.4 mg/ml |
| Other | DAPI mounting medium | Vectashield | H-1200–10 | |
| Commercial assay or kit | Vybrant Multicolor Cell-labelling kit | Thermo Fisher Scientific | Cat.#: V22885, V22886 | 5 µl/ml |
| Commercial assay or kit | Caspase3/CPP32 fluorometric assay kit | Biovision | Cat.#: K105 | |
| Commercial assay or kit | ATP fluorometric assay kit | Sigma-Aldrich | Cat.#: MAK190 | |
| Commercial assay or kit | TruSeq Stranded mRNA kit | Illumina | Cat.#: 20020594 | |
| Commercial assay or kit | Nucleospin RNA extraction kit | Macherey-Nagel | Cat.#: 740,955 | |
| Commercial assay or kit | Sensifast cDNA synthesis kit | Bioline | Cat.#: BIO-65053 | |
| Commercial assay or kit | SensiFAST SYBR NO-ROX kit | Bioline | Cat.#: BIO-98020 | |
| Commercial assay or kit | NK cell isolation kit | Miltenyi Biotec | Cat.#: 130-092-657 | |
| Chemical compound, drug | Alexa Fluor 647 Phalloidin | Invitrogen | A22287 | IF (1/200) |
| Chemical compound, drug | Matrigel | Sigma-Aldrich | 2,222 S; CAS: 22862-76-6 | |
| Chemical compound, drug | Actinomycin D | Sigma-Aldrich | A9415 | |
| Chemical compound, drug | Tetramethyl rhodamine ethyl ester perchlorate (TMRE) | Thermo Fisher Scientific | T669 | |
| Chemical compound, drug | CCCP (carbonyl cyanide 3-chlorophenyl hydrazone) | Sigma-Aldrich | C2759 | |
| Chemical compound, drug | Mitosox | Thermo Fisher Scientific | M36008 | |
| Chemical compound, drug | CellROX Deep red reagent | Thermo Fisher Scientific | C10422 | |
| Chemical compound, drug | Propidium Iodide | Sigma-Aldrich | P4864; CAS: 25535-16-4 | (100 µg/ml) |
| Chemical compound, drug | Ribonuclease A | Sigma-Aldrich | A8950 | |
| Chemical compound, drug | Cycloheximide (CHX) | Sigma-Aldrich | C7698-1G | |

*Appendix 1 Continued on next page*

*Appendix 1 Continued*

| Reagent type (species) or resource | Designation | Source or reference | Identifiers | Additional information |
|---|---|---|---|---|
| Chemical compound, drug | SYTOX Green | Thermo Fisher Scientific | S34860 | |
| Chemical compound, drug | CFSE | Invitrogen CellTrace | C34570 | (1 µl/ml) |
| Chemical compound, drug | Polybrene | Sigma-Aldrich | H9268; CAS: 28728-55-4 | |
| Chemical compound, drug | Blasticidin | invivogen | Cat.#:ant-bl | |
| Antibody | COX IV (rabbit monoclonal) | Cell Signaling | Cat# 4850, RRID:AB_2085424 | IF (1/200) WB (1/1000) |
| Antibody | Cytochrome c (mouse monoclonal) | Cell Signaling | Cat# 12963, RRID:AB_2637072 | IF (1/200) |
| Antibody | Vinculin (mouse monoclonal) | Sigma-Aldrich | Cat#V9131; RRID: AB_477629 | IF (1/400) |
| Antibody | Paxillin (mouse monoclonal) | BD Biosciences | Cat#610052; RRID: AB_397464 | IF (1/500) |
| Antibody | NF-$\kappa$B p65 (mouse monoclonal) | Cell Signaling | Cat# 6956, RRID:AB_10828935 | IF (1/400) |
| Antibody | anti-Mouse IgG (H + L) Highly Cross-Adsorbed Secondary Antibody, Alexa Fluor 647 (donkey polyclonal) | Thermo Fisher Scientific | Cat# A-31571, RRID:AB_162542 | IF (1/300) |
| Antibody | anti-Mouse IgG3 Cross-Adsorbed Secondary Antibody, Alexa Fluor 488 (goat polyclonal) | Thermo Fisher Scientific | Cat# A-21151, RRID:AB_2535784 | IF (1/300) |
| Antibody | Beta Actin (mouse monoclonal) | Sigma-Aldrich | Cat#A3854; RRID: AB_262011 | WB (1/1000) |
| Antibody | PARP1 (rabbit monoclonal) | Cell Signaling | Cat#9,532 S; RRID: AB_659884 | WB (1/1000) |
| Antibody | caspase-3 (rabbit polyclonal) | Cell Signaling | Cat# 9662, RRID:AB_331439 | WB (1/1000) |
| Antibody | GFP (rabbit polyclonal) | Thermo Fisher Scientific | Cat# A-11122, RRID:AB_221569 | WB (1/1000) |
| Antibody | BAX (rabbit polyclonal) | Cell Signaling | Cat# 2772, RRID:AB_10695870 | WB (1/1000) |
| Antibody | Bak (rabbit monoclonal) | Cell Signaling | Cat# 12105, RRID:AB_2716685 | WB (1/1000) |
| Antibody | HSP60 (rabbit polyclonal) | Cell Signaling | Cat# 4870, RRID:AB_2295614 | WB (1/1000) |

*Appendix 1 Continued*

| Reagent type (species) or resource | Designation | Source or reference | Identifiers | Additional information |
|---|---|---|---|---|
| Antibody | K48-linkage Specific Polyubiquitin (rabbit monoclonal) | Cell Signaling | Cat# 8081, RRID:AB_10859893 | WB (1/1000) |
| Antibody | c-IAP1 (rabbit monoclonal) | Cell Signaling | Cat# 7065, RRID:AB_10890862 | WB (1/1000) |
| Antibody | c-IAP2 (rabbit monoclonal) | Cell Signaling | Cat# 3130, RRID:AB_10693298 | WB (1/1000) |
| Antibody | XIAP (rabbit monoclonal) | Cell Signaling | Cat# 14334, RRID:AB_2784533 | WB (1/1000) |
| Antibody | HSC70 (mouse monoclonal) | Santa Cruz Biotechnology | Cat# sc-7298, RRID:AB_627761 | WB (1/1000) |
| Antibody | H3K27ac (rabbit monoclonal) | Diagenode | Cat#: C15210016, RRID:AB_2904604 | WB (1/1000) |
| Antibody | H3K9me3 (mouse monoclonal) | Diagenode | Cat#: C15200153, RRID:AB_2904605 | WB (1/1000) |
| Antibody | BCL-xL (rabbit monoclonal) | Cell Signaling | Cat#2,764 S; RRID: AB_2228008 | WB (1/1000) |
| Antibody | Bcl-2 (mouse monoclonal) | Cell Signaling | Cat# 15071, RRID:AB_2744528 | WB (1/1000) |
| Antibody | MCL1 (rabbit polyclonal) | Cell Signaling | Cat# 4572, RRID:AB_2281980 | WB (1/1000) |
| Antibody | Lamin A/C (mouse monoclonal) | Cell Signaling | Cat# 4777, RRID:AB_10545756 | WB (1/1000) |
| Antibody | anti-rabbit IR Dye 680RD (goat) | LI-COR | Cat# 925–68071, RRID:AB_2721181 | WB (1/10,000) |
| Antibody | anti-mouse IR Dye 800CW (goat) | LI-COR | Cat# 925–32210, RRID:AB_2687825 | WB (1/10,000) |
| Antibody | anti-mouse IR Dye 680RD (goat) | LI-COR | Cat# 926–68070, RRID:AB_10956588 | WB (1/10,000) |
| Antibody | anti-rabbit IR Dye 800CW (goat) | LI-COR | Cat# 926–32211, RRID:AB_621843 | WB (1/10,000) |
| Software, algorithm | ImageJ | NIH | RRID:SCR_003070 | |
| Software, algorithm | Prism v5.0 | https://www.graphpad.com/ | RRID:SCR_002798 | |
| Software, algorithm | Primer-blast | http://www.ncbi.nlm.nih.gov/tools/primer-blast/ | RRID:SCR_003095 | |
| Software, algorithm | edgeR package v3.32.1 | https://doi.org/10.1093/bioinformatics/btp616 | RRID:SCR_012802 | |
| Software, algorithm | Bioconductor | https://www.bioconductor.org/ | RRID:SCR_006442 | |

*Appendix 1 Continued on next page*

*Appendix 1 Continued*

| Reagent type (species) or resource | Designation | Source or reference | Identifiers | Additional information |
|---|---|---|---|---|
| Software, algorithm | Rsubread v2.4.3 | https://www.bioconductor.org/packages/release/bioc/html/Rsubread.html | RRID:SCR_016945 | |
| Software, algorithm | MultiQC | https://multiqc.info/ | RRID:SCR_014982 | |
| Software, algorithm | R version 4.0.3 (2020-10-10) | https://cran.r-project.org/ | RRID:SCR_001905 | |
| Software, algorithm | Rsamtools v2.6.0* | https://bioconductor.org/packages/Rsamtools | RRID:SCR_002105 | |
| Software, algorithm | Analyze of Caliper (AnalyzeDirect) | https://analyzedirect.com/ | RRID:SCR_005988 | |
| Software, algorithm | CaseViewer | https://www.3dhistech.com/caseviewer | RRID:SCR_017654 | |
| Strain, strain background (*Mus musculus*) | NMRI Foxn1 nu/nu | Janvier Labs | SM-NMRNU-F | |
| Strain, strain background (*Escherichia coli*) | NEB 5-alpha competent | New England BioLabs | C2987I | |
| Strain, strain background (*Escherichia coli*) | NEB Stable competent | New England BioLabs | C3040I | |

