## [Editor Report]

The authors provide data supporting the notion that while cells expressing higher levels of cIAP1 or *XIAP* have no confined migratory advantage, these apoptosis inhibitors are upregulated in response to confined migration, which provides cells with a migratory and survival (e.g., anoikis resistance) advantage as well as means to evade NK cells, resulting in increased metastasis.

---

## [Decision Letter]

**Decision letter after peer review:**

[Editors’ note: the authors submitted for reconsideration following the decision after peer review. What follows is the decision letter after the first round of review.]

Thank you for submitting the paper "Confined migration promotes cancer metastasis through resistance to anoikis and increased invasiveness" for consideration by *eLife*. Your article has been reviewed by 3 peer reviewers, and the evaluation has been overseen by a Reviewing Editor and a Senior Editor.

Comments to the Authors:

We are sorry to say that, after consultation with the reviewers, we have decided that this work will not be considered further for publication by *eLife*.

The study proposes that confined migration renders breast cancer cells resistant to apoptosis via NFkappaB-dependent mechanisms. The consensus among the reviewers was that while the technical aspect of the study is impressive and experiments were very well performed, demonstrating the value of mimetic bioengineering approaches, the presented data did not include the necessary rigor to support the postulated central premise.

Specifically, while reviewers agreed that the team conducted a technically impressive study and that the bioengineering aspect of the system is exciting and merits broad use by investigators, there was also consensus regarding the numerous shortcomings of the study. The main concern was that the authors did not address whether the observed effects were "selected" by the provided conditions, or "induced" by the experimental settings. Two of the reviewers noted, and all agreed during the discussion period, that the lack of demonstration to whether confined migration is a causative or a selection process constitutes a major shortcoming. Further, added issues had to do with the incorrect use of statistical approaches, which often rendered biologically similar results as "statistically significant", while other times results that seemed rather biologically distinct were referred to statistically similar or unchanged. It was also noted that the discussion should address recent/similar reported observations made by another group (in Cell). There were some additional points where more data was required to support conclusions, such as whether expression levels of anti-apoptotic proteins are evident prior to or following confined migration (please note detailed provided points by each reviewer). It was hence the consensus that results, statistics, and conclusions need significant clarifications, and it was agreed that the scope of the needed revisions is substantial.

We wish you good luck with this work and we hope you will consider *eLife* for future submissions.

*Reviewer #1:*

This manuscript by Fanfone et al., describes the use an in vitro model of confined migration through transwells via a serum gradient, to study how mechanically challenged breast cancer cells acquire IAP-mediated resistance against anoikis. They demonstrate that these mechanically challenged breast cancer cells survive low-attachment culture to form spheroids. Additionally, the authors demonstrate that after confined migration NFkB is activated but potentially dispensable, as when it is blocked, it does not impact anoikis or spheroid formation. Also, mechanically challenged breast cancer cells demonstrate enhanced motility. Finally, the authors also demonstrate in vivo how mechanically challenged breast cancer cells were more metastatic competent to successfully colonize the lungs. This is an impressive manuscript that reveals how developing an in vitro model of confined migration provides much needed insight as to how microenvironmental constriction can in fact aid cancer cells in acquiring a pro-survival and enhanced migratory phenotype to successfully metastasize.

This is very interesting paper and the authors should be commended on all the work that was put into it. The conclusions were mostly supported by the data, and if some areas could be clarified would be strengthened.

P6 line 194: do you mean cIAP2 over expression was dispensable?

Figure 2g: is there a legend missing?

Suppl Figb is confusing, could you clarify in the legend?

P7 line 223: Change "This" to "To"

P7 line 224: please fix extra paragraph spacing.

P8 line 259: delete "on".

Figure 3i and in the corresponding text: how thick is this Matrigel plug? Was there any cell invasion/migration in the z-plane that could be affecting some of them

P8 line 280 or Figure 3m,n: could you clarify the following: When mechanically challenged multiple times, these cells appear to gain additional migratory capabilities though it was not clear if these serially mechanically challenged cells were migrating on a scratch wound or invading through a Matrigel plug. Also, in the text the authors write that serially mechanically challenged cells gain directionality but there were no data on directionality ratios. Could you extract that from your existing data?

*Reviewer #2:*

The authors present a detailed account of the anti-apoptotic characteristics of breast cancer cells that have undergone confined migration, specifically demonstrating an upregulation of cIAP1, cIAP2 and *XIAP* that promotes anoikis resistance. These cells were also more migratory and potentially displayed a resistance to immune surveillance.

On the whole the paper is technically well performed, although the conclusion that confined migration causes these effects needs further work to be validated. A key question is whether confined migration has caused these changes to occur within individual cells, or whether the assay design results in selection of cells with these characteristics.

For example, it may not be a surprise that selecting cells on the basis of their ability to migrate through a membrane would result in cells with an increased random migratory capacity? Therefore it may also be possible that these innately migratory cells have different expression patterns that include an upregulation of anti-apoptotic proteins? Therefore, there is a need to determine whether these cells were present in the population prior to confined migration, or whether these characteristics were acquired during the process.

On the whole, this manuscript is very well presented and well written. The results and clearly presented, although there is scope for a closer investigation of some of the claims of significance, as well as a need to address the causative vs selective nature of confined migration.

The main thing I would suggest is an experiment that is capable of determining whether the cells with upregulated cIAP1/cIAP2/*XIAP* (or decreased H3K27ac) are present within the population prior to the process of confined migration. If these cells are already present then it doesn't change the validity of the paper, but it completely changes the interpretation. If the cells are not present, and there is a change following confined migration, then the conclusions of the paper stand.

This could be done by endogenously tagging cIAP1/2/X, or by correlative, antibody based single-cell imaging of the two populations using some of the high content approaches already presented in this paper.

*Reviewer #3:*

Mechanical stress has been emerging as the key factor for activating pro-metastatic features, and authors hypothesize confined migration results in anoikis (death in suspension) and increased invasiveness.

By clever repurposing of transwell membranes, authors have generated a confined migration assay (CM), in which cells that have crossed the 3 micron-pore membrane are collected, cultured and further analyzed. The intensity of the CM-related response was shown to increase with number of CM rounds, and the response decays and reverts back 5 days after the CM event.

The resistance to anoikis achieved by CM was not conferred by compressive stress, or migration without constriction through 8-micron pores, and it was further demonstrated to rely on NFkB activation and IAP regulation at post-transcriptional level.

Using RNASeq, authors next show that while transcription is globally inhibited, resulting in lower nuclear stiffness, components of cell adhesion and regulation of NK-cell cytotoxicity were upregulated. Both of these functions were elegantly and succinctly confirmed by time-lapse measurements of cell velocities and immunofluorescence.

Finally, in vivo experiments confirmed metastasis was increased in tail-vein injected cells post-CM.

Major comments:

1. It would be important to demonstrate that, prior to confined migration, cells have similar features, and that CM is not selecting for subpopulation of cells (e.g. low lamin expression).

2. A recent study by Nader et al., published in Cell 2021, notes that breast cancer cells exposed to confinement go through nuclear envelope breakage and TREX1-dependent DNA damage which causes increase in invasiveness. It would be important to describe this study in Discussion, and, if possible, link it to here presented results.

---

## [Author Response]

[Editors’ note: The authors appealed the original decision. What follows is the authors’ response to the first round of review.]

Reviewer #1:This manuscript by Fanfone et al., describes the use an in vitro model of confined migration through transwells via a serum gradient, to study how mechanically challenged breast cancer cells acquire IAP-mediated resistance against anoikis. They demonstrate that these mechanically challenged breast cancer cells survive low-attachment culture to form spheroids. Additionally, the authors demonstrate that after confined migration NFkB is activated but potentially dispensable, as when it is blocked, it does not impact anoikis or spheroid formation. Also, mechanically challenged breast cancer cells demonstrate enhanced motility. Finally, the authors also demonstrate in vivo how mechanically challenged breast cancer cells were more metastatic competent to successfully colonize the lungs. This is an impressive manuscript that reveals how developing an in vitro model of confined migration provides much needed insight as to how microenvironmental constriction can in fact aid cancer cells in acquiring a pro-survival and enhanced migratory phenotype to successfully metastasize.This is very interesting paper and the authors should be commended on all the work that was put into it. The conclusions were mostly supported by the data, and if some areas could be clarified would be strengthened.P6 line 194: do you mean cIAP2 over expression was dispensable?Figure 2g: is there a legend missing?Suppl Figb is confusing, could you clarify in the legend?P7 line 223: Change "This" to "To"P7 line 224: please fix extra paragraph spacing.P8 line 259: delete "on".Figure 3i and in the corresponding text: how thick is this Matrigel plug? Was there any cell invasion/migration in the z-plane

This point is now addressed in the Material and Methods section: since the volume of the matrigel is 50 μL in a 96-well format, we determined that the thickness of the plug is 2.55 mm. As for the potential cell invasion in the z-plane that might bias the analysis, we did not observed any out of focus cells throughout the whole 24 to 48 hours imaging which indicates all cells maintained their z-plane position.

P8 line 280 or Figure 3m,n: could you clarify the following: When mechanically challenged multiple times, these cells appear to gain additional migratory capabilities though it was not clear if these serially mechanically challenged cells were migrating on a scratch wound or invading through a Matrigel plug.

We addressed chemotaxis (transwell assay) in Figure 3M and collective cell migration (wound healing) in Figure 3N. This is now changed in the manuscript.

We thank the reviewer for their supportive comment. Regarding all misspellings and ambiguities, we have now addressed all the above-mentioned points in the revised manuscript.

Also, in the text the authors write that serially mechanically challenged cells gain directionality but there were no data on directionality ratios. Could you extract that from your existing data?

We apologize for misleading the reviewer when we wrote, for example, in line 278:

“…breast cancer cells subjected to a single round of CM did not outperform the control cells when assessed for directional cell migration and invasion (Figure 3F-K)”.

Our intention was to write, “assessed by collective cell migration and invasion”, since these assays are based on collective uni-directional cell migration and invasion, or wound-healing assays (through a matrigel plug). This was now corrected throughout the text.

Reviewer #2:The authors present a detailed account of the anti-apoptotic characteristics of breast cancer cells that have undergone confined migration, specifically demonstrating an upregulation of cIAP1, cIAP2 and XIAP that promotes anoikis resistance. These cells were also more migratory and potentially displayed a resistance to immune surveillance.On the whole the paper is technically well performed, although the conclusion that confined migration causes these effects needs further work to be validated. A key question is whether confined migration has caused these changes to occur within individual cells, or whether the assay design results in selection of cells with these characteristics.For example, it may not be a surprise that selecting cells on the basis of their ability to migrate through a membrane would result in cells with an increased random migratory capacity? Therefore it may also be possible that these innately migratory cells have different expression patterns that include an upregulation of anti-apoptotic proteins? Therefore, there is a need to determine whether these cells were present in the population prior to confined migration, or whether these characteristics were acquired during the process.On the whole, this manuscript is very well presented and well written. The results and clearly presented, although there is scope for a closer investigation of some of the claims of significance, as well as a need to address the causative vs selective nature of confined migration.The main thing I would suggest is an experiment that is capable of determining whether the cells with upregulated cIAP1/cIAP2/XIAP (or decreased H3K27ac) are present within the population prior to the process of confined migration. If these cells are already present then it doesn't change the validity of the paper, but it completely changes the interpretation. If the cells are not present, and there is a change following confined migration, then the conclusions of the paper stand.

We really appreciate the reviewer’s positive remarks regarding our study.

As for addressing the causative versus selective nature on confined migration, we agree this is a very important question that was not fully addressed in the first manuscript. Of note, this a major concern also raised by Reviewer 3.

Now we have performed several experiments that argue in the favor of a causative effect of confined migration. To summarize our reasoning, we found that post-CM there is an upregulation of IAPs when breast cancer cells are grown in anoikis-promoting conditions and that these cells are also more resistant to anoikis.

If these were the effect of a selection mechanism, this would imply that cells with higher levels of IAPs or being more resistant to anoikis (pre-existent in cell population) would have a migration advantage when undergoing CM.

To formally test this, we have done the following experiments:

1. First, we used CRISPR Ctl, cIAP1 and *XIAP* cells (with Ctl considered as cIAP1/*XIAP* high and the CRISPR as low-expressing cells). We then stained the CTL and the CRISPR cells with the lipophilic cyanine dyes DiO (green) and DiI (red), respectively. These cells were mixed at a known ratio (50:50) and subjected to confined migration. At the end, the proportion of Ctl:CRISPR cIAP1 and Ctl:CRISPR *XIAP* was re-assessed and scored, revealing a similar 50:50 ratio.

This implies that cells expressing higher levels of cIAP1 or *XIAP* (the Ctl cells) have no CM advantage and that CM is not selecting for a specific cell population. This is now shown in Figure 2K-L.

2. In a similar manner, we derived single-cell clones from the parental MDA-MB-231 cells and tested the expression of both cIAP1 and *XIAP*. We then selected two clones that have either low (clone 8) or higher (clone 9) expression of both cIAP1 and *XIAP*. We marked these and tested them in CM as described above (clone 8 stained in green with DiO and 9 in red using DiI). The conclusion was similar: a higher expression level of cIAP1 and *XIAP* does not impart an advantage for CM. See Author response image 1.

**Author response image 1. sa2fig1:** 

3. To test whether CM selects cancer cells initially displaying an increased resistance to anoikis, we designed the following experiment: we enriched in MDA-MB-231 cells that are resistant to anoikis by growing them in ultra-low attachment conditions for 7 days. As above, these cells were then marked in red with DiI while parental cells were stained in green with DiO, mixed at 50:50 ratio and then subjected to CM, exactly as described for (1) and (2). Since after CM we found the same 50:50 ratio, this suggests that CM is not selecting for cells displaying resistance to anoikis. This is included in Figure 2M, N.

Reviewer #3:Major comments:1. It would be important to demonstrate that, prior to confined migration, cells have similar features, and that CM is not selecting for subpopulation of cells (e.g. low lamin expression).

We now address the issue of causative versus selective process of CM that was rightly raised by the second reviewer as well. With the experiments presented above (reviewers 2, main comment), we hope to convince R3 that CM is inducing the observed phenotype (increased IAPs and resistance to anoikis). In addition, as shown bellow the breast cancer cells undergoing confined migration have the same expression of lamin A/C as control cell, arguing against a selective pressure to allow low lamin-expressing cells to navigate more efficiently through narrow obstacles (see Figure 3 —figure supplement 1K). We also argue that lamin expression sets a threshold of CM competence. To establish whether lamins have a role during MDA-MB-231 CM, we stably overexpressed lamin A GFP (see Figure 3 —figure supplement 1H) and set up a competition-like scenario between lamin A-GFP overexpressing and control cells stained in red with DiI, similar to the experiments performed in Figure 2K, M (see Figure 3 —figure supplement 1I). Confirming previous studies, we also found that MDA-MB-231 cells with higher levels of lamin A have a lower capacity to perform CM, most probably due to increased nuclear stiffness (Figure 3 —figure supplement 1J).

2. A recent study by Nader et al., published in Cell 2021, notes that breast cancer cells exposed to confinement go through nuclear envelope breakage and TREX1-dependent DNA damage which causes increase in invasiveness. It would be important to describe this study in Discussion, and, if possible, link it to here presented results.

We thank the reviewer for the excellent suggestion to discuss the new study from Nader and colleagues (Sept. 2021, Cell) in the light of our results. This is now discussed as follows:

“These data support the existing relationship between mechanical stress and increased aggressiveness ^9 54^. A recent study from Nader and colleagues elegantly showed that invasive foci in breast cancer are enriched in constricted cells characterized by deformed nuclei and frequent TREX1-dependant DNA lesions. Relevant to our study, these cells were in a partial EMT state that enhanced their invasion potential ^9^.”